# ResMem: Learn what you can and memorize the rest

**Zitong Yang**
Stanford University
Stanford, CA 94305
zitong@berkeley.edu

**Michal Lukasik**
Google Research
New York, NY, 10011
mlukasik@google.com

**Vaishnavh Nagarajan**
Google Research
New York, NY, 10011
vaishnavh@google.com

**Zonglin Li**
Google Research
New York, NY, 10011
lizonglin@google.com

**Ankit Singh Rawat**
Google Research
New York, NY, 10011
ankitsrawat@google.com

**Manzil Zaheer**
Google Research
New York, NY, 10011
manzilzaheer@google.com

**Aditya Krishna Menon**
Google Research
New York, NY, 10011
adityakmenon@google.com

**Sanjiv Kumar**
Google Research
New York, NY, 10011
sanjivk@google.com

## Abstract

The impressive generalization performance of modern neural networks is attributed in part to their ability to *implicitly* memorize complex training patterns. Inspired by this, we explore a novel mechanism to improve model generalization via *explicit* memorization. Specifically, we propose the *residual-memorization* (*ResMem*) algorithm, a new method that augments an existing prediction model (e.g., a neural network) by fitting the model's residuals with a $k$-nearest neighbor based regressor. The final prediction is then the sum of the original model and the fitted residual regressor. By construction, ResMem can explicitly memorize the training labels, even when the base model has low capacity. We start by formulating a stylized linear regression problem and rigorously show that ResMem results in a more favorable test risk over a base linear neural network. Then, we empirically show that ResMem consistently improves the test set generalization of the original prediction model across standard vision and natural language processing benchmarks.

## 1 Introduction

Large neural networks achieve remarkable *generalization* on test samples despite *memorization* of training samples, in the sense of achieving zero training error [54]. Several recent analyses have established that, under certain settings, memorization is *sufficient* to achieve generalization [3, 15, 5, 40, 4], and, more surprisingly, can even be *necessary* [17, 19, 11]. These works suggest that suitable memorization can be a valuable desiderata for learning. While increasing model size is a conceptually simple strategy to enable memorization, this has the obvious downside of significantly increasing the cost of model training and serving. This raises a natural question: *are there alternate mechanisms to improve the memorization (and thus generalization) of a relatively small model?*

In this paper, we propose *residual memorization* (*ResMem*), a simple yet effective mechanism that achieves this goal (cf. Figure 1). Compared to the *implicit* memorization performed by large neural models, the key idea behind ResMem is to perform *explicit* memorization via a separate $k$-nearest neighbor component. Specifically, ResMem involves first training a standard neural network $f_{\mathsf{DeepNet}}$, and then explicitly *memorizing the model's residuals* with a $k$-nearest neighbor based regressor $r_{\mathsf{kNN}}$.

37th Conference on Neural Information Processing Systems (NeurIPS 2023).

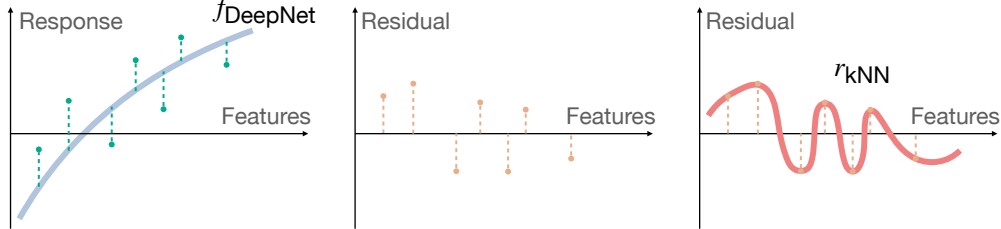

(a) **Step 1:** learn the training set.  (b) **Step 2:** compute the residual.  (c) **Step 3:** memorize the residual.

Figure 1: Illustration of the *residual memorization* (ResMem) algorithm. In a nutshell, we first fit a small deep network $f_{\text{DeepNet}}$ on the training sample (Figure 1(a)). When this network is non-memorizing, it incurs non-zero *residual* errors in its predictions (Figure 1(b)). We then fit a $k$-nearest neighbor based regressor on these residuals (Figure 1(c)). The final prediction is given by the sum of the initial network and $k$-NN regressor predictions. In all three figures, the $x$-axis represents the features in a supervised learning problem. In Figure 1(a), the $y$-axis represents the targets of prediction. In Figure 1(b) and 1(c), the $y$-axis represents the residual of the initial fitting from **Step 1**.

Memorization through $k$-nearest neighbor can be efficiently computed with various approximation schemes (e.g. [24]). Subsequently, the ResMem prediction on an instance $x$ is given by the sum of the two components, i.e., $f_{\text{DeepNet}}(x) + r_{\text{kNN}}(x)$.

We start by formulating a stylized linear regression problem that captures the essence behind ResMem (cf. Section 3). Our analysis (Theorem 3.3) shows that, without ResMem, the test risk of the base linear neural network decreases to an irreducible constant as the number of samples goes to infinity. In contrast, the test risk of ResMem decreases to zero. The insight of theoretical analysis is that ResMem augments the capacity of the parametric linear network by adding a non-parametric component (i.e., nearest-neighbor).

Empirically, we show that such explicit memorization indeed leads to generalization benefits: ResMem consistently improves the test accuracy of a baseline DeepNet on image classification tasks with CIFAR100 [33], and autoregressive language modeling on C4 [42] (Section 4). Towards understanding this improved performance, we hypothesize that ResMem works by learning in two-stages (cf. Section 4.4). Specifically, we posit that the initial DeepNet $f_{\text{DeepNet}}$ learns some *coarse* structure, and ResMem $r_{\text{kNN}}$ supplements the DeepNet prediction with *fine-grained* details (cf. Figure 3). We verify our hypothesis via qualitative analysis on CIFAR100 and C4 (Section 4.4).

To summarize, our contributions are:

(1) We propose *residual-memorization* (*ResMem*), a two-stage learning algorithm that combines a base prediction model with a nearest neighbor regressor (cf. Figure 1);

(2) We theoretically analyze the rate of convergence of ResMem on a stylized linear regression problem, and show that it can improve upon the base prediction model (Section 3).

(3) We empirically demonstrate that ResMem improves test performance of neural networks (cf. Section 4), particularly when the training set is extremely large;

## 1.1 Applicable scenarios of ResMem

From our theoretical and empirical analysis, we posit that ResMem (Figure 1) yields the largest margin of improvement over a base DeepNet when it is infeasible to perform *implicit* memorization with the latter. We discuss three such scenarios below. Each of our main empirical or theoretical results roughly corresponds to one of these settings.

- **Complex dataset.** In this scenario, the Bayes-optimal decision boundary is very complex, and is beyond the capability of the neural network itself. To demonstrate this, we analyze a theoretical linear regression problem where the target regression function is not contained in the hypothesis class of linear neural networks (cf. Section 3).

- **Large sample size.** Here, the number of training samples is large enough to make training set interpolation (i.e., achieving zero training error) infeasible for a given neural network model. For example, current large language models (LLMs) may be trained for at most a single epoch over trillions of examples [12]. By contrast, ResMem can circumvent this issue by explicitly memorizing the training samples. We emulate this scenario by considering a causal language modeling task on the C4 dataset (cf. Section 4.3).

- **Small model.** In many practical settings, one may prefer a smaller model over a state-of-the-art model due to the training and deployment cost constraints. We emulate such a setting through an image classification task where it is indeed feasible to memorize the training data perfectly using state-of-the-art neural networks, but instead, we use smaller neural networks for computational efficiency (cf. Section 4.2).

## 2 Related work

We discuss two types of related work: Section 2.1 for literature on memorization and generalization that motivates the ResMem algorithm; Section 2.2 for other related algorithms similar to ResMem.

### 2.1 Memorization for generalization: prior work

**Memorization is compitable for generalization.** Overparameterized neural models with many more parameters than training samples have the capacity to perfectly fit (or *interpolate*) even random training labels [54]; i.e., they can drive the empirical loss to zero for *any* training set. At the same time, when trained on real-world datasets, increasing model complexity tends to *improve* model performance [40, 52]; that is, the models do not simply memorize the training sample, but rather learn generalizable patterns. Several works have sought to understand the reasons behind this behaviour, both empirically [2] and theoretically [3, 15, 8, 5, 40, 36, 38, 4, 48, 50, 53]. One recurring message from the theory is that memorization (in the form of interpolation) can be sufficient for generalization.

**Memorization can be necessary for generalization.** Some recent works [17, 11] showed that memorization — either in the sense of interpolation, or in a more general sense of stability [18] — may be *necessary* for generalization. Feldman [16] considered a setting where the label distribution exhibits a *long-tailed* distribution, and showed that to prevent incurring a large error on the large number of under-represented classes, it may be necessary to memorize many of their associated training samples. Cheng et al. [11] considered a linear regression setting where it is beneficial to fit the training targets to error lower than the Bayes-error (i.e., the inherent noise in the targets).

### 2.2 Relation to existing algorithms

**Nearest neighbor method.** The $k$-nearest neighbor ($k$-NN) [14, 32, 26, 7] method assigns label to a test sample based on the label of its nearest neighbor(s) in the training set. Owing to its *simplicity*, *flexibility* in defining input similarity, and *computational efficiency* with various approximation schemes [22, 39], this method remains popular. However, the performance of $k$-NN drops as data becomes high dimensional [10, 39]. Therefore, to apply it to high dimensional data such as image and text [55], one approach is to learn a representation of data using neural networks [44]. Following this approach, [13] finds that applying $k$-NN directly to memorize the training labels $y_i$ yields similar performance with the original softmax based neural network classification. In contrast, ResMem applies $k$-NN to memorize the *residual* $r_i$ over the predictions of a base network.

**Boosting and residual fitting.** Boosting algorithms such as AdaBoost [20] seek to construct an ensemble of "weak learner" models with good generalization. AdaBoost achieves this in an iterative manner, and can be interpreted as a particular instantiation of *forward stage-wise regression* [21], a classical procedure from statistics [23, 1, 47]. Intuitively, at each round, one builds a new weaker learner by fitting the residual of the ensemble of weak learners constructed thus far. This fitting is performed iteratively. ResMem can be loosely regarded as a two round boosting algorithm where the first "weak learner" is the base neural network and the second "weak learner" is the nearest-neighbor component. Note that there is no need for the thrid "weak learner", because the nearest-neighbor component already perfectly memorizes the neural network residuals.

**Memory-augmented language models.** In the language modelling literature, several works explore combining neural models with an external database or memory, which can be queried to retrieve additional context [34, 25, 6, 35]. Closer to our work, Khandelwal et al. [30] employ a linear combination of neural network and $k$-NN classifier components. However, a crucial difference is that our $k$-NN components memorizes the *residual* of the DeepNet prediction, whereas Khandelwal et al. [30] memorizes the *target label* directly; i.e., their approach is akin to an ensemble of $k$-NN and a deep network. Various forms of memory have also been considered in generic classification problems [41, 48, 51]. This line of literature again differs from ResMem in that their memory tries to memorize labels directly, whereas ResMem memorizes the *residuals*, leading to a natural combination of the neural network and the memory component.

**Model compression for large neural networks.** Since ResMem boosts the test accuracy of a small, non-memorizing neural network, we can also view it as a technique that allows a small network to match the performance of a larger one. This relates to the model compression literature. Distillation [29, 9] is a popular strategy for compressing a large neural model to a smaller one. For a survey of other effective strategies, including pruning, see Menghani [37]. In Appendix C.2, we discuss how ResMem can be regarded as a "dual procedure" of distillation.

## 3 Theoretical results

As discussed in Section 1.1, ResMem yields the largest improvement when implicit memorization is infeasible. In this section, we formulate (cf. Section 3.1) and analyze (cf. Theorem 3.3) a stylized linear regression problem that concretizes such a setting.

Recall that ResMem (Figure 1) involves first training a base neural network $f_{\mathsf{DeepNet}}$, and then fitting the residual of $f_{\mathsf{DeepNet}}$ on the same training data using a nearest-neighbor regressor $r_{\mathsf{kNN}}$. For feasibility of theoretical analysis, we simplify $f_{\mathsf{DeepNet}}$ with a single layer linear neural network, i.e. linear regression, and we consider 1-nearest neighbor instead of $k$-nearest neighbor to memorize the residual of this network. Our results suggests that ResMem improves test-time generalization by augmenting the capacity of the base model with a non-parametric nearest-neighbor component.

### 3.1 Assumptions and setting

In this section, we present the setup and assumptions for the stylized linear regression problem. We consider a setting where the function class that we minimize over does *not* include the ground-truth function that relates the covariates to the response. Therefore, even with infinite samples, the test loss will decay to a positive constant. We exactly characterize the rate of decay, and show that it converges to 0 under ResMem. Our analysis rests on the following assumptions.

**Assumption 3.1** (Distribution of covariates). The distribution of covariate $\boldsymbol{x} \in \mathbb{R}^d$, denoted by $\mathbb{P}_{\boldsymbol{x}}$, is the uniform distribution[1] over a Euclidean norm ball centered at the origin of radius $\sqrt{d+2}$. The choice of radius ensures that $\mathbb{E}_{\boldsymbol{x} \sim \mathbb{P}_{\boldsymbol{x}}} \boldsymbol{x}\boldsymbol{x}^\mathsf{T} = \boldsymbol{I}$.

**Assumption 3.2** (Linear regression over norm ball). Consider the problem of learning a linear function $f_\star(\boldsymbol{x}) = \langle \boldsymbol{x}, \boldsymbol{\theta}_\star \rangle$ with $\|\boldsymbol{\theta}_\star\| = 1$ from training data $\{(\boldsymbol{x}_i, y_i)\}_{i=1:n}$ where $\boldsymbol{x}_i \overset{\text{i.i.d.}}{\sim} \mathbb{P}_{\boldsymbol{x}}$ and $y_i = f_\star(\boldsymbol{x}_i)$ using the function class

$$\mathcal{F} = \{\boldsymbol{x} \mapsto \langle \boldsymbol{x}, \boldsymbol{\theta} \rangle, \|\boldsymbol{\theta}\| < L\}. \tag{1}$$

We assume $L < 1$ so that the problem belongs to the "hard generalization" scenario discussed in Section 1.1, where the hypothesis space is inadequate to fit the function on its own.

ResMem proceeds by first learning a linear function $f_n(\boldsymbol{x}) = \langle \boldsymbol{\theta}_n, \boldsymbol{x} \rangle$ from $\mathcal{F}$ through empirical risk minimization (ERM):

$$\boldsymbol{\theta}_n = \underset{\|\boldsymbol{\theta}\| \leq L}{\operatorname{argmin}} \frac{1}{n} \sum_{i=1}^n \left[ \langle \boldsymbol{x}_i, \boldsymbol{\theta} \rangle - y_i \right]^2. \tag{2}$$

---

[1]For more general distributions, the theoretical result will depend on quantities like $\mathbb{P}_{\boldsymbol{x}}(\mathcal{B}(\widetilde{\boldsymbol{x}}, h))$, where $\mathcal{B}(\widetilde{\boldsymbol{x}}, h)$ means a ball of radius $h$ that is centered at $\widetilde{\boldsymbol{x}}$. We took uniform distribution for simplicity and to obtain exact dependence on $d$.

The empirical risk minimizer $f_n$ should be thought of as the analog of $f_{\mathsf{DeepNet}}$ in the deep learning context. It defines a ground-truth residual function $r_\star(\boldsymbol{x}) = f_\star(\boldsymbol{x}) - f_n(\boldsymbol{x})$. Now we fix a test covariate $\widetilde{\boldsymbol{x}} \sim \mathbb{P}_x$. ResMem "memorizes" the residual function through the 1-nearest neighbor to $\widetilde{\boldsymbol{x}}$

$$r_n(\widetilde{\boldsymbol{x}}) = r_\star(\widetilde{\boldsymbol{x}}_{(1)}) = f_\star(\widetilde{\boldsymbol{x}}_{(1)}) - f_n(\widetilde{\boldsymbol{x}}_{(1)}), \tag{3}$$

where $\widetilde{\boldsymbol{x}}_{(1)}$ is the nearest neighbor to $\widetilde{\boldsymbol{x}}$ among the training covariates $\boldsymbol{x}_1, \ldots, \boldsymbol{x}_n$:

$$\widetilde{\boldsymbol{x}}_{(1)} = \operatorname*{argmin}_{\boldsymbol{x} \in \{\boldsymbol{x}_1, \ldots, \boldsymbol{x}_n\}} \|\boldsymbol{x} - \widetilde{\boldsymbol{x}}\|.$$

The final prediction is

$$f_n^{\mathsf{ResMem}}(\widetilde{\boldsymbol{x}}) = f_n(\widetilde{\boldsymbol{x}}) + r_n(\widetilde{\boldsymbol{x}}). \tag{4}$$

Observe that if $\widetilde{\boldsymbol{x}}$ coincides with any training sample, $f_n^{\mathsf{ResMem}}(\widetilde{\boldsymbol{x}}) = f_\star(\widetilde{\boldsymbol{x}})$, i.e., we have explicit memorization. Note that we worked with 1-nearest neighbor regressor for simplicity instead of the general $k$-nearest neighbor algorithm. The effect of choosing different $k$ is not the main focus of this theoretical analysis.

## 3.2 A decomposition of the target function

Next, we introduce a decomposition of $f_\star$, which will help us analyze various components that make up the risk. Define

$$\boldsymbol{\theta}_\infty = \operatorname*{argmin}_{\|\boldsymbol{\theta}\| \leq L} \mathbb{E}_{\boldsymbol{x} \sim \mathbb{P}_x} \left[ \langle \boldsymbol{\theta}, \boldsymbol{x} \rangle - \langle \boldsymbol{\theta}_\star, \boldsymbol{x} \rangle \right]^2,$$

$$= \operatorname*{argmin}_{\|\boldsymbol{\theta}\| \leq L} \|\boldsymbol{\theta} - \boldsymbol{\theta}_\star\| = L\boldsymbol{\theta}_\star,$$

which is what ERM learns in the limit of $n \to \infty$. We can think of $\boldsymbol{\theta}_\infty$ as the best function that ERM can learn. Then, we can decompose $\boldsymbol{\theta}_\star$ into $\boldsymbol{\theta}_\star = \boldsymbol{\theta}_\infty + \boldsymbol{\theta}_\perp$, where $\boldsymbol{\theta}_\perp = \boldsymbol{\theta}_\star - \boldsymbol{\theta}_\infty$. This decomposition can be generalized beyond linear regression. Since $\boldsymbol{\theta}_\infty$ defines a function $f_\infty(\boldsymbol{x}) = \langle \boldsymbol{x}, \boldsymbol{\theta}_\infty \rangle$, for general non-linear functions, the argument above can be generalized to the decomposition of $f_\star$ to an learnable and non-learnable part

$$f_\star = f_\infty + f_\perp.$$

Intuitively, $f_\infty$ is the best function in $\mathcal{F}$ that ERM can learn, and $f_\perp$ is beyond the capacity of ERM due to the particular choice of function class. ResMem approximates $f_\perp$ using the non-parametric nearest neighbor method, and therefore expanding the capacity of the original hyphesis class.

## 3.3 A decomposition of the prediction error

We now introduce a decomposition of the prediction risk that reveals how ResMem algorithm boosts generalization. Note that the prediction error of ResMem is

$$\mathbb{E}\left[ \left( f_n^{\mathsf{ResMem}}(\widetilde{\boldsymbol{x}}) - f_\star(\widetilde{\boldsymbol{x}}) \right)^2 \right]. \tag{5}$$

It can be decomposed into two components: $\mathbb{E}\left[ f_n^{\mathsf{ResMem}}(\widetilde{\boldsymbol{x}}) - f_\star(\widetilde{\boldsymbol{x}}) \right]^2 \leq 3 \times$

$$[ \underbrace{\mathbb{E}(f_n(\widetilde{\boldsymbol{x}}) - f_\infty(\widetilde{\boldsymbol{x}}))^2 + \mathbb{E}(f_n(\widetilde{\boldsymbol{x}}_{(1)}) - f_\infty(\widetilde{\boldsymbol{x}}_{(1)}))^2}_{T_1} + \underbrace{\mathbb{E}(f_\infty(\widetilde{\boldsymbol{x}}) - f_\star(\widetilde{\boldsymbol{x}}) - f_\infty(\widetilde{\boldsymbol{x}}_{(1)}) + f_\star(\widetilde{\boldsymbol{x}}_{(1)}))^2}_{T_2} ].$$

We provide the detail of the decomposition in Section B.1. We can see that $T_1$ arises due to the difference between $f_n$ and $f_\infty$ (i.e., the estimation error), which, as we will show later, goes to 0 as $n$ goes to infinity:

$$T_1 \to 0 \text{ as } n \to \infty.$$

On the other hand, $T_2$ arises due to the limited capacity of $\mathcal{F}$. It captures an irreducible error of the risk, which in general is **not** asymptotically zero. However, because of the explicit memorization ResMem algorithm introduces ($\widetilde{\boldsymbol{x}}_{(1)} \to \widetilde{\boldsymbol{x}}$ as $n \to \infty$), we also have

$$T_2 \to 0 \text{ as } n \to \infty.$$

This decomposition provides a statistical perspective on ResMem: it preserves the asymptotic consistency of $T_1$ as in classical learning problems while enforcing the asymptotic consistency of $T_2$ through the nearest-neighbor method.

## 3.4 Main theoretical result

Given the set up above, we are ready to state the main theoretical result of the paper, which characterizes the rate at which test risk of ResMem approaches 0. The proof is in Appendix B.

**Theorem 3.3** (Risk for ResMem algorithm). *For the problem defined in Assumption 3.2 with covariates distribution in Assumption 3.1, the ResMem prediction rule $f_n^{ResMem}(\widetilde{\boldsymbol{x}})$ defined in equation* (4) *achieves risk* (5)

$$\mathbb{E}\left[f_n^{ResMem}(\widetilde{\boldsymbol{x}}) - f_\star(\widetilde{\boldsymbol{x}})\right]^2 \lesssim d^2 L^2 n^{-2/3} + d^2(1-L)^2 \left[\frac{\log\left(n^{1/d}\right)}{n}\right]^{1/d},$$

*where $\lesssim$ denotes inequality up to a universal constant independent of $d, n$ and $L$.*

The result includes contribution from two terms introduced in Section 3.3:

- $T_1 \lesssim d^2 L^2 n^{-2/3}$ that arises due to the difference between $f_n$ and $f_\infty$.
- $T_2 \lesssim \left[\log\left(n^{1/d}\right)/n\right]^{1/d}$ that vanishes as the nearest neighbor of the test point approaches the test point itself $\widetilde{\boldsymbol{x}}_{(1)} \to \widetilde{\boldsymbol{x}}$.

The two terms $T_1$ and $T_2$ can be viewed as "two stages of learning". Without the ResMem memorization component, we have the usual story of machine learning: $T_1 \to 0$ at the usual parametric rate, and $T_2$ stays as an irreducible error, so the overall test error diminishes to a constant at a very fast rate. With the introduction of nearest neighbor memorization procedure, $T_2$ can also be reduced to 0 at a slower rate, whereas the fast decay of $T_1$ is still preserved.

This result shows why it is *not favorable* to use the $k$-nearest neighbor component to memorize the response directly: as a corollary of setting $L = 0$ in Theorem 3.3, pure nearest neighbor would result in an overall slow rate of $\approx n^{-1/d}$. However, with ResMem, we can enjoy benefit of having the test loss being asymptotically 0, while also enjoying the fast rate of $n^{-2/3}$ for smaller sample sizes.

# 4 Empirical results

In this section, we present empirical results on image classification and language modeling that showcase the efficacy of ResMem. In Section 4.1, we present details of applying the ResMem algorithm to classification problems on real dataset. In Section 4.2 and Section 4.3, we present the setup and the result for vision and language experiments, respectively. In Section 4.4 we conduct an empirical analysis to explain where the improved accuracy of ResMem comes from. Finally, in addition to evaluating the improvement ResMem algorithm over DeepNet itself, we compare ResMem with other reasonable baselines including [31] in Appendix F.

## 4.1 Details of ResMem algorithm for classification

We consider multi-class classification problems over instances $\mathcal{X}$ and labels $\mathcal{Y} \doteq \{1, 2, \ldots, L\} = [L]$. Given training examples $S = \{(x_i, y_i)\}_{i \in [n]} \in (\mathcal{X} \times \mathcal{Y})^n$, the goal is to learn a scorer $f : \mathcal{X} \to \mathbb{R}^L$ that, given an instance, assigns an affinity score for each label. Such an $f$ should minimize the *misclassification error* on test samples:

$$L_{01}(f) \doteq \mathbb{P}_{(x,y)}(y \neq \texttt{pred}(f(x))), \tag{6}$$

where $\texttt{pred}(z) \doteq \arg\max_{y' \in [L]} z_{y'}$, and $\mathbb{P}$ is the distribution over labelled instances. To achieve this, one typically minimizes the *empirical loss*

$$\hat{L}_\ell(f) \doteq \frac{1}{n} \sum_{i \in [n]} \ell(y_i, f(x_i)),$$

where $\ell : [L] \times \mathbb{R}^L \to \mathbb{R}_+$ is a loss function. Ideally, one would like to use $\ell_{01}(y, f(x)) \doteq 1(y \neq \texttt{pred}(f(x)))$; for computational tractability, it is popular to instead use a *surrogate loss*, such as the softmax cross-entropy.

Given the notation above, ResMem operates as follows:

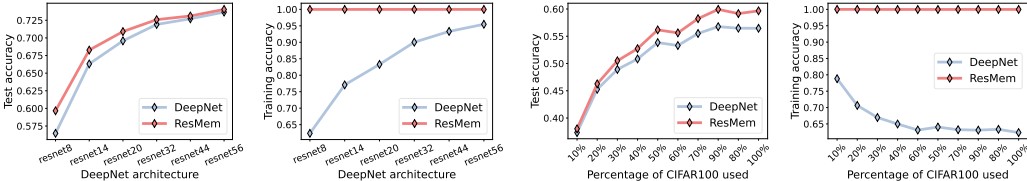

(a) Test(left)/Training (right) v.s. architectures.  (b) Test(left)/Training (right) acc. v.s. sample size.

Figure 2: ResMem improvement on CIFAR100 with respect to training sample size and deep network architecture. **(a):** Using progressively larger CIFAR-ResNet architecture. **(b):** Using $10\%, 20\%, \ldots, 100\%$ of training data.

1. **Train the base DeepNet.** Train a neural network $f_{\mathsf{DeepNet}}$ on the training samples $S$ as usual.

2. **Prepare the residual data.** Compute the *residual* [2] prediction of each training example as

$$r_i = \mathsf{onehot}(y_i) - \mathsf{softmax}(f_{\mathsf{DeepNet}}(x_i)/T), \ \forall \, i \in [n],$$

   where $\mathsf{onehot} \colon \mathcal{Y} \to \mathbb{R}^L$ is the standard encoding that maps the label to a probability vector. Here, $T$ is a hyperparameter corresponding to the "temperature" scaling of the softmax operation. Then, we employ the output of an intermediate layer of the base network $f_{\mathsf{DeepNet}}$, denoted by $z_i = \phi(x_i)$, as an embedding for the training instance $x_i$. These embeddings are utilized for the nearest neighbor search in the next step.

3. **Predict via memorized residuals.** To obtain a prediction on a test sample $\widetilde{x} \in \mathcal{X}$, first compute its embedding $\widetilde{z} = \phi(\widetilde{x})$. Then, use soft $k$-nearest neighbor method to build a function $r_{\mathsf{kNN}}$ defined by weights $\overline{w}_i(\widetilde{x})$:

$$r_{\mathsf{kNN}}(\widetilde{x}) = \sum_{i=1}^{n} \overline{w}_i(\widetilde{x}) \cdot r_i. \tag{7}$$

   The weights $\overline{w}_i(\widetilde{x})$ satisfy $\sum_i \overline{w}_i(\widetilde{x}) = 1$, and are computed from raw weights $w_i$ as follows:

$$w_i = \exp(-\|\widetilde{z} - z_i\|_2/\sigma), \quad \overline{w}_i(\widetilde{x}) \propto \mathbb{1}\left(w_i \geq w_{(k)}\right) w_i,$$

   where $w_{(k)}$ represents the $k$-th largest entry of $w_i$'s. Note that $k$ and $\sigma$ are two hyperparameters that collectively controls the locality of nearest neighbor search.

We make the final prediction based on the following scorer:

$$f_{\mathsf{ResMem}}(\widetilde{x}) = \mathsf{softmax}(f_{\mathsf{DeepNet}}(\widetilde{x})/T) + r_{\mathsf{kNN}}(\widetilde{x}). \tag{8}$$

*Remark* 4.1 (Explicit memorization). Smaller $k$ or $\sigma$ corresponds to putting higher weight on residuals of the closest neighboring training examples. For sufficiently small $k$ and $\sigma$, $f_{\mathsf{ResMem}}$ achieves exact memorization of the training sample, i.e., $\mathsf{pred}(f_{\mathsf{ResMem}}(x_i)) = y_i$ for all $i \in [n]$.

*Remark* 4.2 (Computation cost). The vision experiments have moderate training sample size, so we perform exact nearest neighbor search and discuss the computation cost in Section 4.2. For language experiments, the training sample size is so large that the exact nearest neighbor computation is infesible, so we rely on *approximate* nearest neighbor search discussed in Section 4.2.

### 4.2 Image classification

In this subsection, we mainly consider image classification task on ResNet [27] with CIFAR100 [33] dataset. We provide additional ImageNet [43] results in Apendix D.

**Setup.** We use CIFAR-ResNet-$\{8, 14, 20, 32, 44, 56\}$ as the base DeepNet. For all six DeepNet training, we use SGD with batch size 128, trained for 256 epochs. We use a peak learning rate 0.4, and momentum 0.9. We warm up the learning rate linearly for the first 15 epochs, and decay the learning rate by 0.1 after epochs $\{96, 192, 224\}$. For ResMem, we use the pre-logit layer as the

---

[2]For an overparameterized network that perfectly fits the training sample, the residuals will all be 0. However, we are interested in either smaller networks or extremely large dataset where implicit memorization is infesible.

## Image classification | Language modeling

| | $y^{\text{ResMem}}$ | $y^{\text{DeepNet}}$ | | $y^{\text{ResMem}}$ | $y^{\text{DeepNet}}$ |
|---|---|---|---|---|---|
| | rose | poppy | ...allow for plenty of headroom inside whilst still being less than 2.5m in height. | height | length |
| | cup | plate | Graphic now consists of all cities with greater than 30,000 locals. Acquiring a home in Spain… | home | residence |
| | squirrel | rabbit | Filmed around 7:30-8:30 a.m. on Friday, March 9, 2012. | March | June |
| | butterfly | bee | ...that will not affect the superior quality of your job. That is possible because we understand how to save... | possible | feasible |
| | palm tree | pine tree | ...answer your questions and schedule the initial meeting. We consistently arrive at the scheduled hour... | consistently | always |

Figure 3: Examples from CIFAR100 and C4 test set with the property that **(i)** $y^{\text{ResMem}}$ is correct; **(ii)** $y^{\text{DeepNet}}$ is wrong but close in meaning. We use red to denote the ResMem prediction and blue to denote DeepNet prediction. The DeepNet predictions capture *coarse* structure (e.g., predicting poppy for a sample whose true label is rose), which can be refined by ResMem capturing the remaining *fine-grained* structure.

image embedding, which has dimension 64. For the nearest neighbor search (Step 3, Section 4.1), we define the distance between two images to be the the $\ell_2$ distance between their embeddings. We use $\sigma = 0.7$, $k = 53$, and $T = 1.4$ to compute the weights for the nearest neighbor regressor. We provide the sensitivity analysis of test accuracy against ResMem parameters in Appendix C (cf. Figure 5).

**Results.** The results for CIFAR-ResNet-$\{8, 14, 20, 32, 44, 56\}$ are reported in Figure 2(a). We can see that ResMem boosts the test accuracy of CIFAR-ResNet8 from 56.46% to **59.66%**, which is between the base DeepNet test accuracy for CIFAR-ResNet8 and CIFAR-ResNet14. To access the statistical reliability of the improvement, we repeat the CIFAR-ResNet-8 experiment 5 times over random initialization of DeepNet etc. We and that the average ResMem accuracy is 59% with standard deviation 0.7%, and the average DeepNet accuracy is 56.5% with standard deviation 0.8%.

Computationally, we estimate the CPU latency of a CIFAR-ResNet-8 to be 15.9 ms for a single test image. By contrast, the $k$-NN step takes 4.8 ms for the same test image. To contextualize the latency cost, the total cost of ResMem with ResNet-8 (15.9 ms + 4.8 ms) is lower than the cost of the next-sized model, i.e., ResNet-14 (26.2 ms). Regarding the memory cost, for a batch size of 1 and images of size 32 x 32, a ResNet-8 ( 68K params) requires 2.5MB, while a ResNet-14 ( 128K params) requires 4MB. Embeddings from a ResNet-8 and ResNet-14 are both 64 dimensional. To embed the entire CIFAR100 training set (50K examples) requires 15MB of disk space.

**Varying sample size.** We repeat the above experiment on CIFAR-ResNet-8 with subsets $(10\%, 20\%, \dots, 100\%)$ of CIFAR100 training data (subsampled uniformly across different classes). The size of the index set for nearest-neighbor search is the same as the training set for base neural networks (e.g., model with 10% CIFAR100 data also uses the same 10% data for nearest-neighbor search). On the left (right) of Figure 2(b), we report the test (training) accuracy of ResMem and baseline DeepNet. As a sanity check, we can see that ResMem always achieves perfect training accuracy, and the DeepNet training accuracy decreases as samples increase (since it's harder to fit larger dataset). We can see that ResMem yields *progressively larger margin of improvement when more data is used*. This trend suggests a desirable property of ResMem: in real problems where the dataset is extremely large, ResMem is expected to bring even greater benefit.

### 4.3 Language modeling

**Setup.** For the language experiment, we use a Decoder-Only T5-{small, large} [42] model and C4 [42] dataset. C4 is generated from scraping the internet and commonly used as a pretraining dataset or part of the pretraining mix. We pre-trained the DeepNet on C4 training split with auto-regressive language modeling task. For experimental efficiency, we used 1% of the C4 training split (which corresponds to 1,639 million tokens) as the retrieval database, and extracted last transformer layer's pre-MLP, post-LayerNorm representations as the key embeddings for $k$NN search, and we created the query embeddings using the whole validation split and the same representation location. For each query, we retrieved 50 neighbors with $L_2$ distance using approximate nearest neighbor search algorithm ScaNN [24]. We used the temperature $T = 1$ for the residual computation and $\sigma = 1$ for computing the neighbor weights. The predicted token is the one with highest probability, similar to greedy decoding, and we measured the prediction accuracy to match the vision experiments.

**Results.** On T5-small, ResMem boosts test accuracy from 38.01% to **40.87%**, which is around the accuracy (40.08%) of a T5-base model without ResMem. On T5-large, ResMem boosts the test accuracy from 44.8% to **45.6%**. This demonstrates that with explicit memorization, we may leverage smaller base language models while reaping the performance benefits of large language models. Computationally, as the index set is quite large (1.6 billion tokens), exact k-nearest neighbor search is infeasible. So we use the approximate nearest neighbor search algorithm ScaNN [24] to reduce compute time. Please see Appendix E for details on base model training and data processing.

### 4.4 Where does the improvement come from?

In this section, we identify test samples that contributes to the accuracy improvement of CIFAR100 with CIFAR-ResNet-8 and C4 with T5-small. Let $\mathsf{Gain}_{\mathsf{ResMem}}$ be the difference between the test accuracy of ResMem and baseline DeepNet:

$$\mathsf{Gain}_{\mathsf{ResMem}} = L_{01}(f_{\mathsf{ResMem}}) - L_{01}(f_{\mathsf{DeepNet}}),$$

where $L_{01}$ is the misclassification error as defined in equation (6). We offer a decomposition of $\mathsf{Gain}_{\mathsf{ResMem}}$ that sheds light into the mechanism behind ResMem. For a test set $\{(x_i, y_i)\}_{i=1}^m$, let $y_i^{\mathsf{ResMem}}$ be the ResMem prediction on instance $x_i$ and let $y_i^{\mathsf{DeepNet}}$ be the baseline neural network prediction on $x_i$. When $y_i^{\mathsf{ResMem}} = y_i^{\mathsf{DeepNet}}$, sample $x_i$ does not contribute to $\mathsf{Gain}_{\mathsf{ResMem}}$. When $y_i^{\mathsf{ResMem}} \neq y_i^{\mathsf{DeepNet}}$, this could arise either from the desirable event where the deep network mis-classifies while ResMem classifies correctly; or from the undesirable event where the ResMem misclassifies, while the deep network classifies correctly. These can be summarized by the TPR (*true positive rate*) and FPR (*false positive rate*) respectively:

$$\mathsf{TPR} = \frac{1}{m} \sum_{i=1}^m \mathbb{1}\{y_i^{\mathsf{DeepNet}} \neq y_i \text{ and } y_i^{\mathsf{ResMem}} = y_i\}. \tag{9}$$

$$\mathsf{FPR} = \frac{1}{m} \sum_{i=1}^m \mathbb{1}\{y_i^{\mathsf{DeepNet}} = y_i \text{ and } y_i^{\mathsf{ResMem}} \neq y_i\}. \tag{10}$$

Note that $\mathsf{Gain}_{\mathsf{ResMem}} = \mathsf{TPR} - \mathsf{FPR}$. The decomposition of $\mathsf{Gain}_{\mathsf{ResMem}}$ says that the gain of ResMem came from the TPR samples, provided they outweigh the FPR samples.

On CIFAR-ResNet-8, we find TPR=5.89% and FPR=2.70%, leading to $\mathsf{Gain}_{\mathsf{ResMem}}$=3.19%. On T5-small with C4 validation split, we find TPR=5.37% and FPR=2.44%, leading to $\mathsf{Gain}_{\mathsf{ResMem}}$=2.93%.

**Analysis of TPR samples** Focusing on the test samples where ResMem helps ($y_i = y_i^{\mathsf{ResMem}} \neq y_i^{\mathsf{DeepNet}}$), we identify a common underlying pattern: while the DeepNet makes an incorrect prediction, it still captures some coarse structure. For example, in CIFAR100, one sample has correct label $y_i = y_i^{\mathsf{ResMem}} = \mathtt{rose}$, but the DeepNet predicts $y_i^{\mathsf{DeepNet}} = \mathtt{poppy}$, i.e., the label of a different type of flower. (cf. Figure 3). We find similar behavior for the language modeling task (cf. Figure 3).

This empirical analysis suggests the DeepNet in isolation can already learn some large scale structures, but is unable to make fine-grained distinctions. This is where ResMem helps: *ResMem helps memorize information in the training label that the DeepNet cannot learn.*

**Additional insights from the decomposition.** In this paper, we choose the ResMem hyperparameters that minimizes the test error on the validation set or, equivalently, maximize $\mathsf{Gain}_{\mathsf{ResMem}}$. Inspired by the decomposition of $\mathsf{Gain}_{\mathsf{ResMem}}$, we propose an alternative hyperparameter selection procedure based on the following optimization problem:

$$\mathrm{maximize}_{\mathsf{FPR}(\texttt{hyperparam.})<0.05}\mathsf{TPR}(\texttt{hyperparam.}),$$

which ensures that ResMem modifies the DeepNet predictions in a more conservative manner. In particular, bounding FPR implies that ResMem has minimal impact on the examples where DeepNet already makes correct predictions. At the same time, a higher value of TPR corresponds maximizing the desirable occurrences where ResMem can correct a wrong prediction by DeepNet.

## 5   Discussion and future works

**Joint training of $k$NN and DeepNet.** The current formulation of ResMem builds the base DeepNet and $k$NN components sequentially. Consequently, the DeepNet is trained completely oblivious to the fact that there is a subsequent $k$NN model that will memorize its residuals. A natural direction of future work is to consider the *joint* training of DeepNet and $k$NN, so that the models can dynamically interact during training to determine which portion of label is for DeepNet to learn, and the remaining is for $k$NN to memorize.

To explore the role of training during the first stage, we re-evaluate the CIFAR-ResNet-8 experiment by stopping DeepNet training at different epochs (Table 1). We can see that when the #epoch is small,

Table 1: Comparison of DeepNet and ResMem accuracy over epochs on CIFAR-ResNet-8 experiment.

| #epoch | 128 | 160 | 192 | 224 | 256 |
|---|---|---|---|---|---|
| DeepNet acc. | 34.0% | 56.2% | 55.6% | 57.2% | 56.6% |
| ResMem acc. | 49.3% | 60.2% | 58.6% | 59.2% | 59.5% |

ResMem has a dramatic improvement in accuracy. One of the key roles of the first training phase is to learn good representations of the training data so the nearest neighbor retrieval is performed on more meaningful representations. This simple experiments suggests that the proposed direction has the potential to dramatically reduce the training time of DeepNet – while obtaining similar test accuracy with the help of ResMem.

**Calibration of ResMem.** A potential problem with applying ResMem to classification is *scorer mis-calibration*. The output of the ResMem prediction vector $f_{\mathsf{ResMem}}(x)$ (8) is not guaranteed to lie on the probability simplex. This is not an issue when we only care about the predicted class membership, since we take the argmax of $f_{\mathsf{ResMem}}(x)$. However, this limitation hinders us to access the *confidence* of the ResMem prediction. To remedy this, a possible future work is to consider alternative notions of residual. For example, we can do memorization in the logit space instead of the probability space. Then, the one-hot encoding of the true label may be replaced by class mean when defining the residual.

**Distribution shift.** Finally, ResMem can be a promising approach to tackle test-time covariate shift. The nearest neighbor modifies the prediction of DeepNet based on the training covariate that are closer to the test covariate, making the algorithm more *adaptive* to the specific test covariate [46].

## Acknowledgements

Part of the work is done while Zitong Yang is at Google Research, New York. We would like to thank Chong You, Yu Sun, Yaodong Yu and anonymous reviewers for their feedback on the final draft. Zitong Yang would like to thank Shuangping Li for discussion regarding the proof of Lemma A.1. Zitong Yang would also like to acknowledge the support of Albion Walter Hewlett Stanford Graduate Fellowship.

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

# A  Some concentration results for uniform random variables

In this section, we state some concentration results that are useful for the theoretical analysis in Section 3. Let $\widetilde{\boldsymbol{x}}, \boldsymbol{x}_1, \ldots, \boldsymbol{x}_n \overset{\text{i.i.d.}}{\sim} \text{Unif}(\mathcal{B}_{\boldsymbol{0}, \sqrt{d+2}})$ be i.i.d. samples from the uniform distribution over the Euclidean norm ball of radius $\sqrt{d+2}$ in $\mathbb{R}^d$. Let

$$Z_n = \min_{\boldsymbol{x} \in \{\boldsymbol{x}_1, \boldsymbol{x}_2, \ldots, \boldsymbol{x}_n\}} \|\widetilde{\boldsymbol{x}} - \boldsymbol{x}\|^2. \tag{11}$$

If $n = 1$, $\mathbb{E}Z_1$ is the sum of the variance of each coordinate of $\text{Unif}(\mathcal{B}_{\boldsymbol{0}, \sqrt{d+2}})$. Therefore, $\mathbb{E}Z_n$ provides a generalized measure of concentration. Intuitively, $\mathbb{E}Z_n \to 0$ as $n \to \infty$. The proposition below provides a upper bound on the rate of convergence.

**Lemma A.1** (Nearest Neighbor concentration). *Given the assumptions above*

$$\mathbb{E}Z_n \lesssim d^2 \left[ \frac{\log\left(n^{1/d}\right)}{n} \right]^{1/d}, \tag{12}$$

*where $\lesssim$ means inequality up to an universal constant independent of $d$ and $n$.*

*Proof.* Define

$$\begin{aligned}
\mathcal{E}_1 &= \{Z_n \leq \delta^2\}, \\
\mathcal{E}_2 &= \{\delta \leq \sqrt{d+2} - \|\widetilde{\boldsymbol{x}}\|\}.
\end{aligned} \tag{13}$$

We will compute two probabilities $\mathbb{P}(\mathcal{E}_1|\mathcal{E}_2)$ and $\mathbb{P}(\mathcal{E}_2)$ that will be useful latter.

$$\begin{aligned}
\mathbb{P}(\mathcal{E}_1^c|\mathcal{E}_2) &= \mathbb{P}(Z_n \geq \delta^2|\mathcal{E}_2) = \mathbb{P}(\|\widetilde{\boldsymbol{x}} - \boldsymbol{x}_i\| \geq \delta, \ \forall i|\mathcal{E}_2), \\
&= \mathbb{E}_{\widetilde{\boldsymbol{x}}}\mathbb{P}(\|\widetilde{\boldsymbol{x}} - \boldsymbol{x}_i\| \geq \delta|\mathcal{E}_2, \widetilde{\boldsymbol{x}})^n = \mathbb{E}_{\widetilde{\boldsymbol{x}}}(1 - \mathbb{P}(\|\widetilde{\boldsymbol{x}} - \boldsymbol{x}_i\| \leq \delta|\mathcal{E}_2, \widetilde{\boldsymbol{x}}))^n, \\
&= \mathbb{E}_{\widetilde{\boldsymbol{x}}}\left[1 - \frac{\text{Vol}(\mathcal{B}_{\widetilde{\boldsymbol{x}}, \delta})}{\text{Vol}(\mathcal{B}_{\boldsymbol{0}, \sqrt{d+2}})}\right]^n = \left[1 - \left(\frac{\delta}{\sqrt{d+2}}\right)^d\right]^n, \\
&\leq \exp\left[-n\left(\frac{\delta}{\sqrt{d+2}}\right)^d\right].
\end{aligned} \tag{14}$$

Next, we compute $\mathbb{P}(\mathcal{E}_2)$

$$\mathbb{P}(\mathcal{E}_2) = \mathbb{P}(\|\widetilde{\boldsymbol{x}}\| \leq \sqrt{d+2} - \delta) = \left(\frac{\sqrt{d+2} - \delta}{\sqrt{d+2}}\right)^d = \left(1 - \frac{\delta}{\sqrt{d+2}}\right)^d. \tag{15}$$

We use $\mathcal{E}_1$ and $\mathcal{E}_2$ to compute the following upper bound

$$\begin{aligned}
\mathbb{E}Z_n &= \mathbb{E}(Z_n|\mathcal{E}_1 \cap \mathcal{E}_2)\mathbb{P}(\mathcal{E}_1 \cap \mathcal{E}_2) + \mathbb{E}(Z_n|(\mathcal{E}_1 \cap \mathcal{E}_2)^c)P((\mathcal{E}_1 \cap \mathcal{E}_2)^c), \\
&\leq \delta^2 + (2\sqrt{d+2})^2 \left(1 - \mathbb{P}(\mathcal{E}_1 \cap \mathcal{E}_2)\right), \\
&= \delta^2 + 4(d+2)\left[1 - \mathbb{P}(\mathcal{E}_1|\mathcal{E}_2)\mathbb{P}(\mathcal{E}_2)\right].
\end{aligned} \tag{16}$$

To find an upper bound for $\mathbb{E}Z_n$, we need to find an upper bound for $1 - \mathbb{P}(\mathcal{E}_1|\mathcal{E}_2)\mathbb{P}(\mathcal{E}_2)$.

$$\begin{aligned}
1 - \mathbb{P}(\mathcal{E}_1|\mathcal{E}_2)\mathbb{P}(\mathcal{E}_2) &= 1 - \left[1 - \mathbb{P}(\mathcal{E}_1^c|\mathcal{E}_2)\right]\mathbb{P}(\mathcal{E}_2), \\
&= 1 - \mathbb{P}(\mathcal{E}_2) + \mathbb{P}(\mathcal{E}_1^c|\mathcal{E}_2)\mathbb{P}(\mathcal{E}_2), \\
&\leq 1 - \mathbb{P}(\mathcal{E}_2) + \mathbb{P}(\mathcal{E}_1^c|\mathcal{E}_2).
\end{aligned} \tag{17}$$

Now choose $\delta = \sqrt{d+2}\,n^{-1/d}\left[\log\left(n^{1/d}\right)\right]^{1/d}$.

$$\mathbb{P}(\mathcal{E}_1^c|\mathcal{E}_2) \leq \exp\left[-n\left(\frac{\delta}{\sqrt{d+2}}\right)^d\right] = \exp\left[-nn^{-1}\log\left(n^{1/d}\right)\right] = n^{-1/d}, \tag{18}$$

and

$$\mathbb{P}(\mathcal{E}_2) = \left(1 - \frac{\delta}{\sqrt{d+2}}\right)^d \geq 1 - d\frac{\delta}{\sqrt{d+2}} = 1 - dn^{-1/d}\left[\log\left(n^{1/d}\right)\right]^{1/d}. \tag{19}$$

Thus

$$1 - \mathbb{P}(\mathcal{E}_1|\mathcal{E}_2)\mathbb{P}(\mathcal{E}_2) \leq 1 - 1 + dn^{-1/d}\left[\log\left(n^{1/d}\right)\right]^{1/d} + n^{-1/d} \lesssim dn^{-1/d}\left[\log\left(n^{1/d}\right)\right]^{1/d}. \tag{20}$$

Combining everything together, we get

$$
\begin{aligned}
\mathbb{E}Z_n &\leq (d+2)n^{-2/d}\left[\log\left(n^{1/d}\right)\right]^{2/d} + 4(d+2) \times dn^{-1/d}\left[\log\left(n^{1/d}\right)\right]^{1/d}, \\
&\lesssim d^2 n^{-1/d}\left[\log\left(n^{1/d}\right)\right]^{1/d}, \\
&= d^2\left[\frac{\log\left(n^{1/d}\right)}{n}\right]^{1/d}.
\end{aligned}
\tag{21}
$$

This completes the proof. $\qquad\square$

**Proposition A.2** ([49] Corollary 6.20). *Let* $\boldsymbol{x}_i \stackrel{\text{i.i.d.}}{\sim} \text{Unif}\left(\mathcal{B}_{\boldsymbol{0},\sqrt{d+2}}\right)$ *for* $i = 1,\ldots,n$ *be uniformly distributed over a ball of radius* $B$ *in* $\mathbb{R}^d$ *centered at* $\boldsymbol{0}$. *Let*

$$\boldsymbol{\Sigma}_n = \frac{1}{n}\sum_{i=1}^n \boldsymbol{x}_i\boldsymbol{x}_i^\mathsf{T}$$

*be the sample covariance matrix. Then*

$$\mathbb{P}(\|\boldsymbol{\Sigma}_n - \boldsymbol{I}\|_{\text{op}} > \varepsilon) \leq 2d\exp\left[-\frac{n\varepsilon^2}{2(d+2)(1+\varepsilon)}\right].$$

# B Proof of Theorem 3.3

In this section, we present the proof of Theorem 3.3. In Section B.1, we provide the detail of the decomposition of the risk into $T_1$ and $T_2$. Then in Section B.2 we compute an upper bound for $T_1$, and compute an upper bound for $T_2$ in Section B.3. Finally, we combine everything together in Section B.4 and completes the proof.

## B.1 Decomposition of the test risk

$$
\begin{aligned}
\mathbb{E}\left[f^{\mathsf{ResMem}}(\widetilde{\boldsymbol{x}}) - f_\star(\widetilde{\boldsymbol{x}})\right]^2 &= \mathbb{E}\left[f_n(\widetilde{\boldsymbol{x}}) + r_n(\widetilde{\boldsymbol{x}}) - f_\star(\widetilde{\boldsymbol{x}})\right]^2, \\
&= \mathbb{E}\left[f_n(\widetilde{\boldsymbol{x}}) - f_\star(\widetilde{\boldsymbol{x}}) - f_n(\widetilde{\boldsymbol{x}}_{(1)}) + f_\star(\widetilde{\boldsymbol{x}}_{(1)})\right]^2, \\
&= \mathbb{E}\left[f_n(\widetilde{\boldsymbol{x}}) - f_\infty(\widetilde{\boldsymbol{x}}) + f_\infty(\widetilde{\boldsymbol{x}}) - f_\star(\widetilde{\boldsymbol{x}}) - f_n(\widetilde{\boldsymbol{x}}_{(1)}) + f_\infty(\widetilde{\boldsymbol{x}}_{(1)}) - f_\infty(\widetilde{\boldsymbol{x}}_{(1)}) + f_\star(\widetilde{\boldsymbol{x}}_{(1)})\right]^2, \\
&\leq 3 \times \underbrace{\left[\mathbb{E}(f_n(\widetilde{\boldsymbol{x}}) - f_\infty(\widetilde{\boldsymbol{x}}))^2 + \mathbb{E}(f_n(\widetilde{\boldsymbol{x}}_{(1)}) - f_\infty(\widetilde{\boldsymbol{x}}_{(1)}))^2\right.}_{T_1} + \underbrace{\mathbb{E}(f_\infty(\widetilde{\boldsymbol{x}}) - f_\star(\widetilde{\boldsymbol{x}}) - f_\infty(\widetilde{\boldsymbol{x}}_{(1)}) + f_\star(\widetilde{\boldsymbol{x}}_{(1)}))^2}_{T_2}\Big],
\end{aligned}
$$
(22)

where in the last inequality, we used the fact that $(a + b + c)^2 < 3(a^2 + b^2 + c^2)$ for any $a, b, c \in \mathbb{R}$.

## B.2 Upper bound on $T_1$.

Since $\mathbb{P}_{\boldsymbol{x}} = \mathrm{Unif}(\mathcal{B}_{\boldsymbol{0}, B})$, we apply the bound $\|\widetilde{\boldsymbol{x}}\|, \|\widetilde{\boldsymbol{x}}_{(1)}\| \leq B$ to obtain

$$
\begin{aligned}
T_1 &= \mathbb{E}[f_n(\widetilde{\boldsymbol{x}}) - f_\infty(\widetilde{\boldsymbol{x}})]^2 + \mathbb{E}[f_n(\widetilde{\boldsymbol{x}}_{(1)}) - f_\infty(\widetilde{\boldsymbol{x}}_{(1)})]^2, \\
&= \mathbb{E}\langle \boldsymbol{\theta}_n - \boldsymbol{\theta}_\infty, \widetilde{\boldsymbol{x}}\rangle^2 + \mathbb{E}\langle \boldsymbol{\theta}_n - \boldsymbol{\theta}_\infty, \widetilde{\boldsymbol{x}}_{(1)}\rangle^2, \\
&\leq \mathbb{E}\|\boldsymbol{\theta}_n - \boldsymbol{\theta}_\infty\|^2 \|\widetilde{\boldsymbol{x}}\|^2 + \mathbb{E}\|\boldsymbol{\theta}_n - \boldsymbol{\theta}_\infty\|^2 \|\widetilde{\boldsymbol{x}}_{(1)}\|^2, \\
&\leq 2B^2 \mathbb{E}\|\boldsymbol{\theta}_n - \boldsymbol{\theta}_\infty\|^2.
\end{aligned}
$$
(23)

As $n$ gets large, the empirical covariance matrix $\boldsymbol{\Sigma}_n = \boldsymbol{X}^\mathsf{T}\boldsymbol{X}/n$ is concentrated around its mean $\boldsymbol{I}$. Let $\boldsymbol{\Delta}_n = \boldsymbol{I} - \boldsymbol{\Sigma}_n$ denote this deviation. For some $\varepsilon \in (0, 1)$, define the following "good event" over the randomness in $\boldsymbol{\Sigma}_n$

$$
\mathcal{A} = \{\|\boldsymbol{\Delta}_n\|_{\mathrm{op}} < \varepsilon\},
$$
(24)

where $\|\boldsymbol{\Delta}_n\|_{\mathrm{op}}$ denotes the operator norm of the deviation matrix. The high level idea of the proof is to condition on the event $\mathcal{A}$ and deduce and upper bound of $\|\boldsymbol{\theta}_n - \boldsymbol{\theta}_\infty\|$ in terms of $\varepsilon$. Then, we use the fact that $\mathcal{A}$ happens with high probability.

Recall that $\boldsymbol{\theta}_\infty = L\boldsymbol{\theta}_\star$, and

$$
\boldsymbol{\theta}_n = \operatorname*{argmin}_{\|\boldsymbol{\theta}\| \leq L} \frac{1}{n}\|\boldsymbol{X}\boldsymbol{\theta} - \boldsymbol{y}\|^2.
$$
(25)

Since $\boldsymbol{y} = \boldsymbol{X}\boldsymbol{\theta}_\star$ by definition, the Lagrangian of the convex program above is

$$
\mathcal{L}(\boldsymbol{\theta}, \lambda) = \frac{1}{n}\|\boldsymbol{X}\boldsymbol{\theta} - \boldsymbol{X}\boldsymbol{\theta}_\star\|^2 + \lambda(\|\boldsymbol{\theta}\|^2 - L).
$$
(26)

The KKT condition suggests that the primal-dual optimal pair $(\boldsymbol{\theta}_n, \lambda_n)$ is given by

$$
\begin{aligned}
\|\boldsymbol{\theta}_n\| &\leq L, \\
\lambda_n &\geq 0, \\
\lambda_n(\|\boldsymbol{\theta}_n\| - L) &= 0,
\end{aligned}
$$
(27)

and at optimality

$$
\begin{aligned}
\nabla_{\boldsymbol{\theta}}\mathcal{L}(\boldsymbol{\theta}_n, \lambda_n) = 0 &\iff \frac{2}{n}\boldsymbol{X}^\mathsf{T}\boldsymbol{X}(\boldsymbol{\theta} - \boldsymbol{\theta}_\star) + 2\lambda_n\boldsymbol{\theta} = 0, \\
&\iff \boldsymbol{\theta}_n = (\boldsymbol{\Sigma}_n + \lambda_n\boldsymbol{I})^{-1}\boldsymbol{\Sigma}_n\boldsymbol{\theta}_\star.
\end{aligned}
$$
(28)

The complementary slackness condition $\lambda_n(\|\boldsymbol{\theta}_n\| - L) = 0$ suggests that either $\lambda_n = 0$ or $\|\boldsymbol{\theta}_n\| = L$. But if $\lambda_n = 0$, the stationary condition $\nabla_{\boldsymbol{\theta}} \mathcal{L}(\boldsymbol{\theta}, \lambda) = 0$ would suggest that $\boldsymbol{\theta}_n = \boldsymbol{\Sigma}_n^{-1} \boldsymbol{\Sigma}_n \boldsymbol{\theta}_\star = \boldsymbol{\theta}_\star \Rightarrow \|\boldsymbol{\theta}_n\| = 1 > L$, a contradiction. (Note that here $\boldsymbol{\Sigma}_n$ is invertible condition on the event $\mathcal{A}$.) Therefore, we must have $\|\boldsymbol{\theta}_n\| = L$. As a result, the primal and dual pair $(\boldsymbol{\theta}_n, \lambda_n)$ is determined by the system of equations

$$\begin{cases} \boldsymbol{\theta}_n &= (\boldsymbol{\Sigma}_n + \lambda_n \boldsymbol{I})^{-1} \boldsymbol{\Sigma}_n \boldsymbol{\theta}_\star, \\ \|\boldsymbol{\theta}_n\| &= L, \\ \lambda_n &> 0. \end{cases} \tag{29}$$

Next, we proceed to compute the deviation $\|\boldsymbol{\theta}_n - \boldsymbol{\theta}_\infty\|$.

$$\begin{aligned} \boldsymbol{\theta}_n &= [(\lambda_n + 1)\boldsymbol{I} - \boldsymbol{\Delta}_n]^{-1} \boldsymbol{\Sigma}_n \boldsymbol{\theta}_\star, \\ &= (\lambda_n + 1)^{-1} \left[ \boldsymbol{I} - \frac{\boldsymbol{\Delta}_n}{\lambda_n + 1} \right]^{-1} \boldsymbol{\Sigma}_n \boldsymbol{\theta}_\star, \\ &= (\lambda_n + 1)^{-1} \left[ \boldsymbol{I} + \sum_{k=1}^{\infty} \frac{\boldsymbol{\Delta}_n^k}{(\lambda_n + 1)^k} \right] (\boldsymbol{I} - \boldsymbol{\Delta}_n) \boldsymbol{\theta}_\star, \\ &= (\lambda_n + 1)^{-1} \left[ \boldsymbol{I} + \sum_{k=1}^{\infty} \frac{\boldsymbol{\Delta}_n^k}{(\lambda_n + 1)^k} - \boldsymbol{\Delta}_n - \sum_{k=1}^{\infty} \frac{\boldsymbol{\Delta}_n^{k+1}}{(\lambda_n + 1)^k} \right] \boldsymbol{\theta}_\star, \\ &= (\lambda_n + 1)^{-1} \boldsymbol{\theta}_\star + (\lambda_n + 1)^{-1} \boldsymbol{\Delta}_n \left[ \sum_{k=1}^{\infty} \frac{\boldsymbol{\Delta}_n^{k-1}}{(\lambda_n + 1)^k} - \boldsymbol{I} - \sum_{k=1}^{\infty} \frac{\boldsymbol{\Delta}_n^k}{(\lambda_n + 1)^k} \right] \boldsymbol{\theta}_\star, \\ &= (\lambda_n + 1)^{-1} \boldsymbol{\theta}_\star + (\lambda_n + 1)^{-1} \boldsymbol{\Delta}_n \left[ \sum_{k=1}^{\infty} \frac{\boldsymbol{\Delta}_n^{k-1} - \boldsymbol{\Delta}_n^k}{(\lambda_n + 1)^k} - \boldsymbol{I} \right] \boldsymbol{\theta}_\star. \end{aligned} \tag{30}$$

Define

$$\boldsymbol{D}_n = \boldsymbol{\Delta}_n \left[ \sum_{k=1}^{\infty} \frac{\boldsymbol{\Delta}_n^{k-1} - \boldsymbol{\Delta}_n^k}{(\lambda_n + 1)^k} - \boldsymbol{I} \right]. \tag{31}$$

Then $\boldsymbol{\theta}_n = (\lambda_n + 1)^{-1} \boldsymbol{\theta}_\star + (\lambda_n + 1)^{-1} \boldsymbol{D}_n \boldsymbol{\theta}_\star$, and

$$\begin{aligned} \|\boldsymbol{D}_n\| &\leq \|\boldsymbol{\Delta}_n\| \left[ 1 + \sum_{k=1}^{\infty} \frac{\|\boldsymbol{\Delta}_n\|^{k-1} + \|\boldsymbol{\Delta}_n\|^k}{(\lambda_n + 1)^k} \right], \\ &\leq \varepsilon \left[ 1 + 2(1 + \lambda_n)^{-1} \sum_{k=1}^{\infty} \left( \frac{\varepsilon}{1 + \lambda_n} \right)^k \right], \\ &= \varepsilon \left( 1 + \frac{2}{1 + \lambda_n} \frac{1}{1 - \frac{\varepsilon}{1+\lambda_n}} \right) \leq 3\varepsilon. \end{aligned} \tag{32}$$

Therefore

$$\begin{aligned} L &= \|\boldsymbol{\theta}_n\|^2 = (\lambda_n + 1)^{-2} + (\lambda_n + 1)^{-2} \boldsymbol{\theta}_\star^{\mathsf{T}} \boldsymbol{D}_n^{\mathsf{T}} \boldsymbol{D}_n \boldsymbol{\theta}_\star + 2(\lambda_n + 1)^{-2} \boldsymbol{\theta}_\star \boldsymbol{D}_n \boldsymbol{\theta}_\star, \\ &\Rightarrow (\lambda_n + 1)^2 L^2 = 1 + \delta_n, \ \delta_n = \boldsymbol{\theta}_\star^{\mathsf{T}} \boldsymbol{D}_n^{\mathsf{T}} \boldsymbol{D}_n \boldsymbol{\theta}_\star + 2 \boldsymbol{\theta}_\star^{\mathsf{T}} \boldsymbol{D}_n \boldsymbol{\theta}_\star. \end{aligned} \tag{33}$$

We can obtain the following bound for $\delta_n$:

$$|\delta_n| \leq \|\boldsymbol{\theta}_\star\|^2 \|\boldsymbol{D}_n\|^2 + 2\|\boldsymbol{\theta}_\star\|^2 \|\boldsymbol{D}_n\| \leq 9\varepsilon^2 + 6\varepsilon \leq 15\varepsilon. \tag{34}$$

Since $1 - \delta_n/2 \leq \sqrt{1 + \delta_n} \leq 1 + \delta_n/2$, and $|\delta_n| \leq 15\varepsilon$, we obtain

$$|(\lambda_n + 1)L - 1| \leq \frac{15\varepsilon}{2} \Rightarrow \left| L - (\lambda_n + 1)^{-1} \right| \leq \frac{15\varepsilon}{2}(\lambda_n + 1)^{-1} \leq \frac{15\varepsilon}{2}, \tag{35}$$

where the last inequality follows as we have $\lambda_n > 0$. Finally,

$$
\begin{aligned}
\boldsymbol{\theta}_n - \boldsymbol{\theta}_\infty &= (\lambda_n + 1)^{-1}\boldsymbol{\theta}_\star - L\boldsymbol{\theta}_\star + (\lambda_n + 1)^{-1}\boldsymbol{D}_n\boldsymbol{\theta}_\star, \\
\Rightarrow \|\boldsymbol{\theta}_n - \boldsymbol{\theta}_\infty\|^2 &= [(1+\lambda_n)^{-1} - L]^2 + (1+\lambda_n)^{-2}\boldsymbol{\theta}_\star\boldsymbol{D}_n^\mathsf{T}\boldsymbol{D}_n\boldsymbol{\theta}_\star + 2(\lambda_n+1)^{-1}[(1+\lambda_n)^{-1} - L]\boldsymbol{\theta}_\star\boldsymbol{D}_n\boldsymbol{\theta}_\star, \\
&\leq 64\varepsilon^2 + 9\varepsilon^2 + 45\varepsilon^2 = 118\varepsilon^2, \\
\Rightarrow \|\boldsymbol{\theta}_n - \boldsymbol{\theta}_\infty\|^2 &\lesssim \varepsilon^2.
\end{aligned}
\tag{36}
$$

Combine the above result with Proposition A.2, we get that

$$
\begin{aligned}
\mathbb{E}\|\boldsymbol{\theta}_n - \boldsymbol{\theta}_\infty\|^2 &= \mathbb{E}(\|\boldsymbol{\theta}_n - \boldsymbol{\theta}_\infty\|^2|\mathcal{A})\mathbb{P}(\mathcal{A}) + \mathbb{E}(\|\boldsymbol{\theta}_n - \boldsymbol{\theta}_\infty\|^2|\mathcal{A}^c)\mathbb{P}(\mathcal{A}^c), \\
&\leq \varepsilon^2 + 4L^2 \times 4d\exp\left[-\frac{n\varepsilon^2}{2(d+2)(1+\varepsilon)}\right],
\end{aligned}
\tag{37}
$$

If we choose $\varepsilon = n^{-1/3}$, we get

$$
\mathbb{E}\|\boldsymbol{\theta}_n - \boldsymbol{\theta}_\infty\|^2 \lesssim dL^2 n^{-2/3},
\tag{38}
$$

which implies that

$$
T_1 \lesssim d^2 L^2 n^{-2/3}.
\tag{39}
$$

## B.3 Upper bound on $T_2$.

Plugging in the formula for $f_\perp(\widetilde{\boldsymbol{x}}) = f_\star(\widetilde{\boldsymbol{x}}) - f_\infty(\widetilde{\boldsymbol{x}}) = \langle\widetilde{\boldsymbol{x}}, \boldsymbol{\theta}_\perp\rangle$, we get

$$
\begin{aligned}
T_2 &= \mathbb{E}[f_\perp(\widetilde{\boldsymbol{x}}_{(1)}) - f_\perp(\widetilde{\boldsymbol{x}})]^2, \\
&= \mathbb{E}\langle\boldsymbol{\theta}_\perp, \widetilde{\boldsymbol{x}}_{(1)} - \widetilde{\boldsymbol{x}}\rangle^2, \\
&\leq (1-L)^2\|\boldsymbol{\theta}_\star\|^2\mathbb{E}\|\widetilde{\boldsymbol{x}} - \widetilde{\boldsymbol{x}}_{(1)}\|^2, \\
&= (1-L)^2\mathbb{E}\|\widetilde{\boldsymbol{x}} - \widetilde{\boldsymbol{x}}_{(1)}\|^2,
\end{aligned}
\tag{40}
$$

where in the last inequality, we used the relation that $\boldsymbol{\theta}_\perp = (1-L)\boldsymbol{\theta}_\star$. Proposition A.1 suggests that

$$
\mathbb{E}\|\widetilde{\boldsymbol{x}} - \widetilde{\boldsymbol{x}}_{(1)}\|^2 \lesssim d^2\left[\frac{\log\left(n^{1/d}\right)}{n}\right]^{1/d},
\tag{41}
$$

which implies

$$
T_2 \lesssim d^2(1-L)^2\left[\frac{\log\left(n^{1/d}\right)}{n}\right]^{1/d}.
\tag{42}
$$

*Remark* B.1 (Comparison with pure nearest neighbor and ERM). If we rely solely on nearest neighbor method, the prediction error is

$$
\mathbb{E}[f_\star(\widetilde{\boldsymbol{x}}) - f_\star(\widetilde{\boldsymbol{x}}_{(1)})]^2 = \mathbb{E}\langle\widetilde{\boldsymbol{x}} - \widetilde{\boldsymbol{x}}_{(1)}, \boldsymbol{\theta}_\star\rangle^2 \leq \mathbb{E}\|\widetilde{\boldsymbol{x}} - \widetilde{\boldsymbol{x}}_{(1)}\|^2.
\tag{43}
$$

On the other hand, if we solely rely on ERM, even with infinite sample, we get

$$
\mathbb{E}[f_\star(\widetilde{\boldsymbol{x}}) - f_\infty(\widetilde{\boldsymbol{x}})]^2 = \mathbb{E}\langle\widetilde{\boldsymbol{x}}, \boldsymbol{\theta}_\star - \boldsymbol{\theta}_\infty\rangle^2 \leq (1-L)^2\mathbb{E}\|\widetilde{\boldsymbol{x}}\|^2.
\tag{44}
$$

We can see from the upper bound that ResMem takes advantage of both

- Projecting $f_\star$ onto $f_\infty$, so that the dependence on the prediction function is reduced from 1 to $(1-L)^2$.

- Memorizing the residuals using nearest neighbor, so that the variance is reduced from $\mathbb{E}\|\widetilde{\boldsymbol{x}}\|^2$ to $\mathbb{E}\|\widetilde{\boldsymbol{x}}_{(1)} - \widetilde{\boldsymbol{x}}\|^2$.

### B.4 Test loss for ResMem.

If we combine the previous two parts together, we get

$$\mathbb{E}\left[\hat{f}(\widetilde{\boldsymbol{x}}) - f_\star(\widetilde{\boldsymbol{x}})\right]^2 \lesssim d^2 L^2 n^{-2/3} + d^2(1-L)^2 \left[\frac{\log\left(n^{1/d}\right)}{n}\right]^{1/d}. \tag{45}$$

This completes the proof of Theorem 3.3.

# C Additional CIFAR100 Results

This section includes additional experiment results on applying ResMem to CIFAR100 dataset.

## C.1 Additional robustness results

In addition to the results already presented in Section 4.2, we also evaluate ResMem performance for each architecture in CIFAR-ResNet{8, 14, 20, 32, 44, 56} and each subset (10%, 20%, ..., 100%) of CIFAR100 training data. We use the same training hyperparameter and the ResMem hyperparameter as described in Section 4.2. Generally, we see that ResMem yields larger improvement over the baseline DeepNet when the network is small and dataset is large.

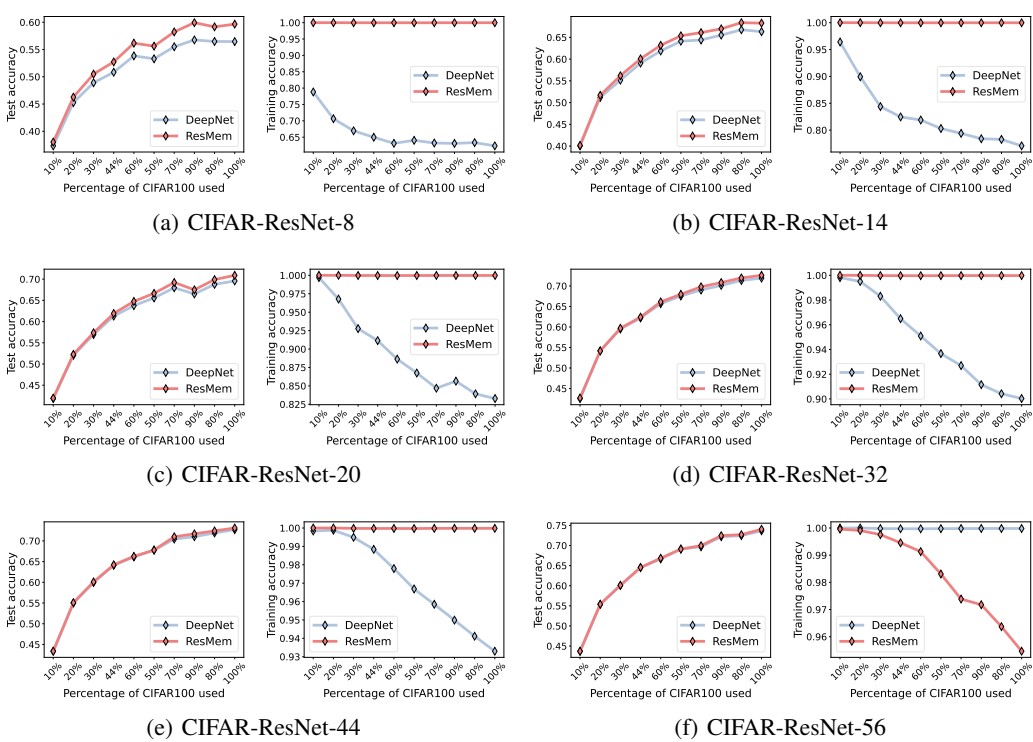

Figure 4: Test(left)/Training (right) accuracy for different sample sizes.

## C.2 Sensitivity analysis for CIFAR100

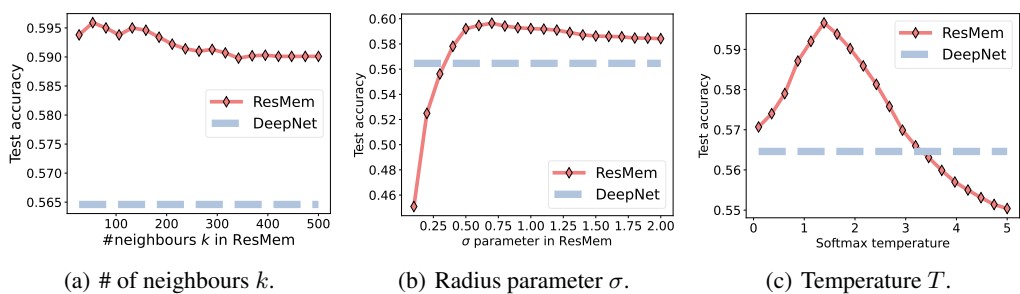

(a) # of neighbours $k$.  (b) Radius parameter $\sigma$.  (c) Temperature $T$.

Figure 5: Sensitivity analysis of ResMem hyperparameters. The $y$-axis represents the CIFAR100 test accuracies, and the $x$-axis represents the sweeping of respective hyperparameters.

**Varying locality parameter $k$ and $\sigma$.** We vary the number of neighbours from $k = 27$ to $k = 500$. We find that ResMem test accuracy is relatively stable across the choice of the number of neighbours (cf. Figure 5(a)). The trend of the curve suggests that as $k \to \infty$, the ResMem test accuracy seems to be converging to a constant level. For $\sigma$, we explored different values of $\sigma \in (0.1, 2.0)$. We observe that the test accuracy has a unimodal shape as a function of $\sigma$, suggesting that there is an optimal choice of $\sigma$ (cf. Figure 5(b)).

**Varying temperature $T$ and connection to distillation.** We tried $T = 0.1$ to $T = 5$, and also identified an unimodal shape for the test accuracy (Figure 5(c)). The fact that we can use different temperatures for (a) training the network and (b) constructing the $k$-NN predictor reminds us of the well-established knowledge distillation procedure [28]. In knowledge distillation, we first use one model (the teacher network) to generate targets at a higher temperature, and then train a second model (the student network) using the *combination* of the true labels and the output of the first network.

ResMem operates in a reversed direction: Here we have a second model (kNN) that learns the *difference* between true labels and the output of the first model. In both cases, we can tune the temperature of the first model to control how much information is retained. This connection offers an alternative perspective that regards ResMem as a "dual procedure" to knowledge distillation.

## D ResMem on ImageNet

This section includes additional experiment results on applying ResMem to ImageNet dataset.

**ImageNet.** In addition to CIFAR100, we also evaluate the performance of ResMem on ImageNet [43]. We employ a family of pre-trained MobileNet-V2 models [45] from Keras[3], with varying widths controlled by a multiplier $a$. For ResMem, we again use the second last layer of DeepNet as a 1280-dimensional embedding of an image and rely on the $\ell_2$ distance between the embeddings for nearest neighbor search (Step 3, Section 4.1). We specify the ResMem parameter of $(k, \sigma, T)$ in the table below. We repeat the experiment over several MobileNet-V2 architectures, with MobileNet-V2-a0.35 being the smallest model and MobileNet-V2-a1.3 being the largest one.

Table 2: Test accuracy for ResMem and baseline deep network for ImageNet data.

| Architecture | ResMem param. | | | Test accuracy | |
|---|---|---|---|---|---|
| | $k$ | $\sigma$ | $T$ | DeepNet | ResMem |
| MobileNet-V2-a0.35 | 10 | 0.6 | 0.4 | 60.2% | **61.2%** |
| MobileNet-V2-a0.5 | 10 | 0.6 | 0.4 | 65.3% | **66.1%** |
| MobileNet-V2-a0.75 | 10 | 0.8 | 0.6 | 69.6% | **70.1%** |
| MobileNet-V2-a1.0 | 20 | 0.4 | 0.4 | 71.3% | **71.8%** |
| MobileNet-V2-a1.3 | 30 | 0.4 | 0.4 | 74.7% | **75.1%** |

We can see that (c.f. Table 2) ResMem boosts the test accuracy by $1\%$ on the smallest model and by $0.4\%$ on the largest model.

## E Additional details of NLP experiments

The Decoder-Only model used in our experiments is essentially the normal Encoder-Decoder architecture with Encoder and Cross-Attention removed. We pretrained both the T5-small and T5-base model on C4 [42] dataset with auto-regressive language modeling task for 1,000,000 steps, with dropout rate of 0.1 and batch size of 128. The learning rate for the first 10,000 steps is fixed to 0.01 and the rest steps follow a square root decay schedule.

During the inference for retrieval key, query embeddings and residuals, we ensured every token has at least 64 preceding context by adopting a sliding window strategy, where a window of 256 token slides from the beginning to the end on each of the articles, with a stride of $256 - 64 = 192$.

---

[3]`https://keras.io/api/applications/mobilenet/`

For residuals, we only stored the top 128 residuals measured by the absolute magnitude, as the residual vector is as large as T5 vocabulary size (i.e., 32128), and storing all 32128 residuals for each token is too demanding for storage. However, when weight-combining the residuals, we zero filled the missing residuals so that all the residual vectors have 32128 elements.

# F    Comparison with other algorithms

We mainly compare ResMem against [31], where the algorithm uses $k$NN to retrive labels directly instead of the residual of the label. In their algorithm, a key aparameter is $\lambda \in [0, 1]$ which specifeis how much weight to give to the neural network and how much for the $k$NN component. In the extreme case of $\lambda$=1, their algorithm reduces to using $k$NN to memorize data directly.

For the language modeling task, we use the C4 dataset and T5-large architecture. As we change the weight [31, Equation (3)] of the DeepNet component, we find the best performing kNN-LM methods has accuracy 44.88% which is lower accuracy than the ResMem accuracy 45.55%. In particular, we obtain the table below

Table 3: Test accuracy for kNN-LM (ResMem accuracy 45.55%)

| kNN weight $\Lambda$ | 0 | 0.2 | 0.4 | 0.5 | 0.6 | 0.8 | 1 |
|---|---|---|---|---|---|---|---|
| kNN-LM accuracy | 44.76% | 44.88% | 44.83% | 44.66% | 44.27% | 42.97% | 40.95% |
| ResMem acc. - kNN-LM acc. | 0.79% | 0.67% | 0.72% | 0.89% | 1.28% | 2.58% | 4.60% |

For image classification with CIFAR-ResNet-8, we run the simple baseline of using $k$-nearest neighbor to directly memorize the labels . We observe the performance: we observe that pure DeepNet has accuracy 56.46%; pure $k$NN memorization has accuracy 54.44%; and ResMem has accuracy 59.66%.

