# OpenReview forum: "ResMem: Learn what you can and memorize the rest"
_NeurIPS.cc/2023/Conference — NeurIPS 2023 poster_

### Official Review · Reviewer_Kq4J · 2023-06-30

**Soundness:** 3 good
**Presentation:** 4 excellent
**Contribution:** 3 good
**Rating:** 6
**Confidence:** 3

**Summary:**

Several recent works investigated the role of memorization in Deep Neural Networks as long as its interplay with generalization capabilities. In this context, this work proposes a two-stage procedure where a standard prediction model is augmented  via a  k-NN based mechanism that explicitly memorizes the model residuals. The authors carried on an interesting theoretical analysis as long as an empirical evaluation of the proposed approach.

**Strengths:**

The proposed two-stage method is a novel and original approach to leverage memorization as a tool for improving neural network performances. The papaer is well written and structured. The theoretical claims are well supported by the empirical findings and their presentation is easy to follow.  I appreciated the clarirty in the theoretical claims exposition and the broad experimental setup.

**Weaknesses:**

I have few remarks with the proposed approach. While reading the paper, the first doubt that comes to mind is why the authors chose to devise a two-stage process instead of the joint training of DeepNet and kNN, as noticed in the Future Work section. Moreover, from similar consideration begs a question on the role of the first training stage: what is the role of the duration of the first training phase? What happens when the second stage is applied too early or too late? (if the first stage comprehends less/more epochs)

Finally, I have some concerns regarding the statistical reliability of the proposed solution: from the current text it is not clear whether the reported results are the average over several runs (with different weights initialization) or not

**Questions:**

1) See "Weaknesses" section: what is the role of the duration of the first training phase? What happens when the second stage based on kNN is applied too early or too late?

2) I have some concerns regarding the statistical reliability of the proposed solution: from the current text it is not clear whether the reported results are the average over several runs (with different weights initialization) or not -- given also that in bigger datasets such as ImageNet the performance improvement is very limited

3) Line 252: from the current text, the parameters seem cherry picked (the ones achieving the best performance on the Test set, from Appendix C). The authors should clarify this.

4) The memory (mainly, givent that the kNN need to keep track of training data)/computational burden of the proposed method should be better discussed (enriching lines 257-260)

5) From the current text, it is not clear whether Assumption 3.1 is easy to be fullfilled or not.

**Limitations:**

Limitations have been adequately addressed.

---

> ### Author Rebuttal · Authors · 2023-08-04
>
> We thank the reviewer for seeing the originality of the ResMem algorithm. We address your questions below.
>
>
> > Joint training of kNN and DeepNet. Role of the duration of the first training phase? What happens when the second stage is applied too early or too late?
>
> This is a very insightful question. We run the suggested experiment on CIFAR100 and CIFAR-ResNet-8 with different training epochs.
>
> | #epoch| 128| 160| 192| 224| 256|
> |---|---|---|---|---|---|
> | DeepNet acc. | 34.0\% | 56.2\% | 55.6\% | 57.2\% |56.6\% |
> | ResMem acc.  | 49.3\% | 60.2\% | 58.6\%  | 59.2\% |59.5\% |
>
> We can see that we the #epoch is small, ResMem has a dramatic improvement in accuracy. One of the key roles of the first training phase is to learn good representations of the training data so the nearest neighbor retrieval is performed on more meaningful representations. From the experiment, it seems that just from the first half of the training stage, the ResNet already picks up some good representation of the data, and it's enough to make the nearest neighbor retrieval meaningful.
>
> As for the joint training of kNN and DeepNet, as pointed out by the reviewer, we have left as future work. The intuition behind two-stage design is from the classical boosting algorithm, where the residual of the previous model is used to fit the current model. ResMem is like one step boosting since there is no residual left after kNN memorization is applied. We believe that both designs of ResMem, joint training or two stage learning, are worth studying. The focus of this paper is on two-stage learning and we left joint training as future work.
>
> > Statistical reliability and poor ImageNet improvement.
>
> We acknowledge that the reported results are not averaged over several runs of random seeds. For CIFAR100 and NLP C4 this is less of an issue, as the margin of improvement is relatively large. We agree for that ImageNet results reported in the Appendix of main body pdf has a small margin of improvement, so statistical reliability comes into play. As for the ImageNet result, we'd like to bring your attention to our ImageNet results in the separate Supplementary PDF we uploaded, where we evaluate ResMem with a different architecture (MobileNet), showing a larger margin of improvement. We also cite the them below:
>
> | Architecture|$k$ |$\sigma$|$T$ |DeepNet test acc.|ResMem test acc.|
> |---|---|---|---|--|---|
> | MobileNet-V2-a0.35|10 | 0.6 | 0.4 | 60.2\% | 61.2\%   |
> | MobileNet-V2-a0.5|10 | 0.6 | 0.4 | 65.3\% | 66.1\% |
> | MobileNet-V2-a0.75|10 | 0.8 | 0.6 | 69.6\% |70.1\% |
> | MobileNet-V2-a1.0|20 | 0.4 | 0.4 | 71.3\% | 71.8\% |
> | MobileNet-V2-a1.3|30 | 0.4 | 0.4 | 74.7\% | 75.1\% |
>
> We can see that ResMem boosts the test accuracy by $1\%$ on the smallest model and by $0.4\%$ on the largest model. We will update the CIFAR100, ImageNet result with confidence interval in the camera ready version. The C4 experiment is more expensive to run, if time permitting we will also report an average improvement. Otherwise we will add a remark in the text clarifying that the result comes from a single run of experiment.
>
> > The memory (mainly, givent that the kNN need to keep track of training data)/computational burden of the proposed method should be better discussed (enriching lines 257-260)
>
> For a batch size of 1 and images of size 32 x 32, a ResNet\-8 (\~68K params) requires 2.5MB, while a ResNet-14 (\~128K params) requires 4MB. Embeddings from a ResNet\-8 and ResNet\-14 are both 64 dimensional. To embed the entire CIFAR100 training set (50K examples) requires ~15MB of disk space. We will add this discussion on top of the current line 257-260.
>
> >  Line 252: from the current text, the parameters seem cherry picked
>
> We acknowledge that the ResMem parameters are not selected strictly following the train/validation split. We use CIFAR-ResNet-8 on the test set to select one combination of hyperparameters. That choice is then fixed throughout the CIFAR experiment across different architecture and dataset size. In that sense, our intention is just to use the test set of get a rough range of numerical values hyperparameters should take. To access the effect of test set tuning, we conduct the following experiment on CIFAR-ResNet-8
>
> We split the 60K (10K+50K) CIFAR100 images into three folds:
>
> * Training Set
> * Validation Set
> * Test Set
>
> Then, we train (run SGD, building the retrieval set) the model $M$ on the Training Set, and make two comparisons. We fix a grid of hyper-param $\mathcal{H}$. We find the optimal hyper-parameter $h^\star\in\mathcal{H}$ that maximizes the accuracy on  *(a): Test Set* and *(b): Validation Set*, and report the accuracy of $M$ with $h^\star$ on the Test Set. Under setting (a) we get 59.65\% accuracy and 59.51\% under (b). This difference suggests that the accuracy reported in the paper is stable under hyper-parameter selection based on test or validation set.
>
> > From the current text, it is not clear whether Assumption 3.1 is easy to be fullìlled or not.
>
> The uniform distribution is chosen to simplify the nearest neighbor analysis for a cleaner result. The radius is chosen to ensure that the covariance matrix is identity. For a more general distribution, we can have an upper bound on the test error involving the term $\mathbb{P}\_\mathbf{X}(\mathcal{B}(x, h))$, where $\mathbb{P}\_X$ is the measure for the covariate distribution and $\mathcal{B}(x, h)$ is a ball of radius $h$ centered at $x$, as in the Theorem 3.1.1 of [1]. The advantage of choosing a specific distribution is that we can see the precise scaling with respect to $d$ and $n$. We will add a sentence explaining the logic behind the choice of covariate distribution in the camera ready version.
>
> [1] George H. Chen; Devavrat Shah, Explaining the success of nearest neighbor methods in prediction, 2018. URL: https://ieeexplore.ieee.org/document/8384208

---

> > ### Comment · Reviewer_Kq4J · 2023-08-14
> >
> > I read the rebuttal and I thank the authors for their answers.  I think that the analysis on *Joint training of kNN and DeepNet*, *memory burden* and *Assumption 3.1* improved the paper contributions.
> >
> > Conversely, I have some concerns regarding both the 1) statistical analys  and the  2) hyperparameter selection procedure.
> >
> > 1) Also the improvements in the case of MobileNet seem to not be very large. However, I understand that the rebuttal time was very small to run many experiments. If possible, the authors could add the confidence intervals before the end of the authors/reviewer discussion phase.
> > 2) The procedure is a bit convoluted but understandable. The competitors hyperparameters were chosen following a comparable approach?
> >
> > Given the current rebuttal, I am convinced by the authors response and I confirm my  **Weak Accept** rating.

---

> > > ### Author Response · Authors · 2023-08-16
> > >
> > > We thank the reviewer for carefully reviewing the paper and our rebuttal. Here is our response regarding the potential concerns.
> > >
> > > > Re. statistical analysis:
> > >
> > >
> > > Note that ResMem is a deterministic algorithm given a base prediction model, so all randomness comes from the base model. In our case, it's the random initialization, training, etc of the neural network, as noted by the reviewer. The MobileNet-V2 we use are pre-trained on ImageNet and is directly loaded from `Keras`, so there is no randomness left. *We are working  towards training our own MobileNet to get the confidence interval, hopefully before the discussion period ends.*
> > >
> > > As a surrogate before the new results, we evaluate the the CIFAR-ResNet-8 experiment over 5 runs, we find that
> > >
> > > * ResMem Test Acc.: `mean = 59%, std = 0.7%`.
> > > * Base ResNet Test Acc.: `mean = 56.5%, std=0.8%`.
> > >
> > > We can see that the standard deviation of ResMem basically inherits the standard deviation of the neural network itself. In this case, the margin of improvement is `2.5%`, which is about `3.5` times standard deviation.
> > >
> > > > The procedure is a bit convoluted but understandable. The competitors hyperparameters were chosen following a comparable approach?
> > >
> > > Yes, both approach follow the same grid search of hyperparameters. Approach (a) did the grid search over the Test Set, and approach (b) searched over the Validation Set. Our experiments shows that the test accuracy in both approaches are comparable.

---

> > > > ### Author Response · Authors · 2023-08-21
> > > >
> > > > An update regarding our efforts to estimate the variation of results on ImageNet. As noted, our primary results use pre-trained models, on top of which ResMem produces deterministic output. To assess sensitivity to variation in the model training, we trained our own MobileNet-v2 models (a = 0.35) on ImageNet. Over 5 trials, we obtain:
> > > >
> > > > DeepNet: mean 53.9, stdev 0.5
> > > >
> > > > ResMem: mean 55.6, stdev 0.4
> > > >
> > > > The gap between ResMem and DeepNet is thus beyond that which could be explained by noise.
> > > >
> > > > As a qualifier, the performance of our DeepNet (53.9) does not reach that of the pre-trained Keras model (60.2). Unfortunately, we couldn't find documentation on the precise training schedule used to produce the pre-trained Keras models [1]; thus, while we made a best attempt to derive a DeepNet with reasonable performance, there is still a gap in the accuracy. Nonetheless, the above indicates that the gain of ResMem is fairly consistent across trials; as we do not expect better trained MobileNets to show _higher_ accuracy variation than the setting above, we hope this provides some evidence for the statistical question on ImageNet.
> > > >
> > > > [1] https://keras.io/api/applications/mobilenet/#mobilenetv2-function

---

### Official Review · Reviewer_diiC · 2023-07-05

**Soundness:** 3 good
**Presentation:** 2 fair
**Contribution:** 2 fair
**Rating:** 5
**Confidence:** 4

**Summary:**

The paper proposes ResMem (residual memorization) to improve neural network predictions by explicitly memorizing training residual patterns. The final prediction is computed by summing the neural network's output and the fitted residual obtained by KNN. The paper provides theories proving that ResMem yields favorable test risk.  Image classification and language modelling experiments demonstrate that ResMem outperforms the original prediction model.

**Strengths:**

- The idea of using KNN to memorize the residual is novel and interesting
- There is a risk bound proof


**Weaknesses:**

- The assumption of the proof (linear function, k=1) is not practical and not aligned with the experiment
- The experiment should compare with approaches that use KNN to store the response/label (e.g. paper [30] cited by the authors should be compared in the NLP task, and please compare with deep KNN paper (Papernot et al., 2018)  in the image classification task )
- The current DeepNet is weak, especially in CIFAR100. The test accuracy is low (<73%) compared to the current SOTA such as Vision Transformer (Dosovitskiy et al., 2020). The DeepNet cannot fit the training dataset (did you tune the model properly and let it train long enough?). Fig. 2a shows that the performance gain reduces as the DeepNet gets bigger. It raises the concern that ResMem may not scale well with big/SOTA backbones and will be not useful in practice.

Papernot, Nicolas, and Patrick McDaniel. "Deep k-nearest neighbors: Towards confident, interpretable and robust deep learning." arXiv preprint arXiv:1803.04765 (2018).

Dosovitskiy, Alexey, Lucas Beyer, Alexander Kolesnikov, Dirk Weissenborn, Xiaohua Zhai, Thomas Unterthiner, Mostafa Dehghani et al. "An image is worth 16x16 words: Transformers for image recognition at scale." arXiv preprint arXiv:2010.11929 (2020).

**Questions:**

- The setup in Sec.4.2: Why do you set up this way?
- Fig. 5: the results seem insensitive to K. Can you explain?

**Limitations:**

No section on Limitations. One potential limitation is the practicality of ResMem. It is unclear if this research direction can lead to better results with strong deep networks.

---

> ### Author Rebuttal · Authors · 2023-08-04
>
> > The assumption of the proof (linear function, k=1) is not practical and not aligned with the experiment
>
> We agree that the toy experiment with linear function and 1-nearest neighbor is very different from the actual empirical. The main purpose of the theory is to design an abstract setting that captures how the residual memorization component can "expand" the capacity of the underlying ERM function class while preserving the fast rate of ERM (lines 201 - 204).
>
> Extension to more complex models are possible.
> * When the exact minimizer $\mathbf{\theta}_n$ is infeasible in closed form (e.g. non-convex objective, model via iterative procedure such as SGD),  one may augment the analysis to additionally consider the effect of optimization error [1] in such cases.
> * For the number of neighbors $k>1$, we can follow the procedure outlined in the section 3.1.1 [2]. The high level logic is that given a test data $\tilde{x}$, there are two bad events that happens with low probability. First, the variance component of kNN regressor is too large. This can be controlled by setting $k$ sufficient large. Second, the neighbors used to compute the average is too far away from $\tilde{x}$. This can be controlled by setting the number of samples $n$ sufficient large relative to $k$. As a result, when $n$ is sufficient large, and $k$ is neither too big or too small, we are able to obtain an upper bound in the regression error (Theorem 3.3.1 of [2]).
>
> Integrating this line of attack to the ResMem setting (where we are memorizing residuals but not labels) would pose additional technical challenges. We believe that the current simplified theory already captures the intuition we wish to convey. We will add a remark to the theory section to discuss further extensions to more complex models in the camera ready version.
>
> [1] Leon Bottou and Olivier Bousquet. The tradeoffs of large scale learning. Advances in Neural Information Processing Systems, volume 20, 2007. URL https://proceedings.neurips.cc/paper/2007/file/0d3180d672e08b4c5312dcdafdf6ef36-Paper.pdf.
>
> [2] George H. Chen; Devavrat Shah, Explaining the Success of Nearest Neighbor Methods in Prediction, 2018. URL: https://ieeexplore.ieee.org/document/8384208
>
>
> > Approaches that use KNN to store the response/label (e.g. paper [30] cited by the authors) should be compared in the NLP task, and please compare with deep KNN paper in the image classification task.
>
> We appreciate the comment regarding the comparative methods. For the NLP task we compared ResMem against the kNN-LM methods from paper [30]. We use the C4 dataset and T5-large architecture. As we change the weight ([equation (3), 30]) of the DeepNet component, we find the best performing kNN-LM methods has accuracy 44.89\% which is lower accuracy than the ResMem accuracy 45.55\%.
>
>
> | kNN weight $\lambda$                    | 0 (pure DeepNet) | 0.1     | 0.2     | 0.3     | 0.4     | 0.5     | 0.6     | 0.7     | 0.8     | 0.9     | 1 (pure kNN) |
> |---|----|---|---|---|---|---|---------|---------|---------|---------|--------------|
> | kNN-LM accuracy| 44.76\%| 44.84\% | 44.88\% | 44.89\% | 44.83\% | 44.66\% | 44.27\% | 43.73\% | 42.97\% | 41.98\% |      40.95%        |
> | ResMem acc. - kNN-LM acc. |0.79\% |  0.71\% |  0.67\% |  0.66\% |  0.72\% |  0.89\% |  1.28\% |  1.82\% |  2.58\% |  3.57\% |      4.60%        |
>
> For image classification, we run the simple baseline of using k-nearest neighbor to directly memorize the labels $y$. We observe the performance: [pure DeepNet: 56.46%; pure kNN memorization: 54.44%; ResMem: 59.66%]. We will add those result to the camera-ready version of the paper.
>
> > The current DeepNet is weak and raises the concern that ResMem may not scale well with big/SOTA backbones and will be not useful in practice.
>
> We believe that ResMem is most effective *when the dataset size is much larger than the model size*, so there are two ends of the spectrum we can go:
> * (a) Fix the dataset size, and evaluate the smaller models;
> * (b) Study problems which have (essentially) unlimited data, and evaluate the larger models.
>
> | Regime | Task                 | Data size      | Model size      | Test accuracy improvement |
> |--------|----------------------|----------------|-----------------|---------------------------|
> | (a)    | Image classification | 50K (CIFAR100) | 67K (ResNet-8)  | 3.2%                      |
> | (b)    | Language modeling    | 33B (C4)       | 30M (T5-small)  | 2.9%                      |
>
> For both (a) and (b), the data size is much larger relative to the model size. The difference is that in (b), the language modeling task, the data size can be very large, so we can also evaluate large models (with 30M parameters). However, for image classification, the size of the dataset is limited by the effort of human laborers. As a result, we *intentionally* use small models to conform with the “data>>model” story.
>
> > The setup in Sec.4.2: Why do you set up this way? Did you tune the model properly and let it train long enough?
>
> The CIFAR-ResNet8 (#param=67K) is a small model for CIFAR100. We follow a standard procedure from [3] with setup details outlined in Section 4.2. Our approach was to reuse a standard approach to optimization, instead of developing one ourselves
>
> [3] Kaiming He, Xiangyu Zhang, Shaoqing Ren and Jian Sun. Deep residual learning for image recognition," CVPR 2016
>
> > Fig. 5: the results seem insensitive to K. Can you explain?
>
> The number of neighbors used in the figure are k = [1,   27,  54,  80, ... , 421, 447, 474, 500]. So there is a big jump in test accuracy from k=0 (no ResMem) to k=1, and the test error becomes insensitive to $k$ for large $k$. One possible explanation is by looking at Theorem 3.3.1 from [2]. They find that the in vanilla kNN (no ResMem), the test error $\varepsilon$ scales as $\varepsilon\sim\frac{1}{k^2}$  w.r.t. to $k$. So from $k=0$ to $k=1$, there is a big jump, and for large $k$, the sensitivity declines.

---

### Official Review · Reviewer_M8UD · 2023-07-06

**Soundness:** 3 good
**Presentation:** 3 good
**Contribution:** 3 good
**Rating:** 6
**Confidence:** 3

**Summary:**

This paper proposes a new learning algorithm, "ResMem," that combines deep neural networks (DNNs) with k-Nearest Neighbors (kNNs). The algorithm is a form of two-stage boosting: we first train a neural network. Then we construct a kNN algorithm that is trained to predict the residuals. The final output combines these two predictions. The nearest-neighbor search is performed in the feature space defined by an intermediate layer of the neural network. The intuition behind this algorithm is the growing body of literature which attributes some of deep learning's success to memorization: why not make it an explicit part of the pipeline?

The first result in the paper is theoretical, giving a toy problem where (a constrained but natural algorithm for) linear regression performs poorly, vanilla kNN also performs poorly, but ResMem does reasonably well.

The remaining results are empirical, applying their algorithm to image classification and language modeling. They show that ResMem provides modest but consistent and significant improvements over the deep networks alone.

**Strengths:**

I think this is a nice idea. As they acknowledge, prior work has explicitly explored connections between kNNs and DNNs, but this paper aims to use the kNN to boost where the DNN fails.

The algorithm is practical and simple, it seems quite reasonable to fit into existing pipelines.

Beyond practice, I think this is also a paper to spark discussion. It provides a complement to the work of Cohen, Sapiro, and Giryes (2018), who used kNNs to probe the memorization capabilities of DNNs.

I felt the introduction and related work sections were well-written.

**Weaknesses:**

One might use ResMem to better understand deep networks, perhaps following Sections 3 and 4.4. Instead, the paper is more like a proposal for a practical algorithm. However, I am not sure where ResMem is the best algorithm available. In Section 1.1, "Applicable Scenarios," the paper gives three ideal settings. The first is "complex dataset," where the task is beyond the capacity of the learner. We only see this setting theoretically in the paper. The other two examples are "large sample size" and "[s]mall model," but in both of these cases it is not clear to me that ResMem makes practical sense. ResMem requires storing representations of the entire training set, so when the sample size is large or the model is small I expect this would be serious overhead.

I found it hard to find the algorithm in the paper: Section 3 analyzes "ResMem for regression," which is described in text and equations. Section 4.1 gives the algorithm for classification in a enumerated list, plus some additional details in the surrounding text. I strongly prefer putting the algorithm in a separate environment, making it clear to the reader what you propose.

The paragraph "Memorization is sufficient for generalization" (line 76) seems extremely confused. I have not read all the cited papers, but my understanding is that they focus on how memorization is *compatible* with generalization. It seems to me that memorization alone *suffices* for generalization only in trivial learning tasks. Concretely, consider an algorithm that uses a lookup table on it training data and answers randomly on unseen points. This always memorizes but only generalizes when the training data captures most of the probability mass of the data distribution.

**Questions:**

Lines 82-83 say "One recurring message from the theory is that memorization (in the form of interpolation) can be sufficient for generalization." Can you connect this statement more directly with an example or two from the preceeding citations?

How do you pick the intermediate layer to use as the representations for the kNN?

ResMem requires storing representations of the training data. Do you think this is an obstacle to its practical application?

Can you present some comparisons about space usage? For instance, in in line 257, you compare the computation of ResNet-14 to that of ResMem (on ResNet-8). What is the footprint of these two models?

**Limitations:**

Aside from potential issues around space usage, which are addressed above, I feel the authors adequately address the limitations of their work.

---

> ### Author Rebuttal · Authors · 2023-08-03
>
> Thanks for seeing the strength of the proposal and providing insightful comments! We will answer your questions below:
>
> > A separate algorithm section.
>
> We agree this is a good idea. The dilemma was that ResMem is a very general algorithm and can be applied to any supervised learning problem (regression/classification). As a result, nailing it to a specific classification or regression setting can potentially hide the generality of the algorithm. We will add a separate algorithm section/box that highlights the three steps in Figure 1 as the main "ResMem Algorithm". We really appreciate this suggestion!
>
> > Discussion on related work regarding "memorization is sufficient for generalization".
>
> We acknowledge that the title of this paragraph is not mathematically precise. Note that here, by "memorization" we mean "the training data is fit to zero error". Indeed, it's not true that _for any_ learning algorithm and data distribution, fitting the training data to zero error is sufficient for generalizing to test data. Rather, we intended to convey the idea that training _neural networks_ to fit _real-world_ training data as perfectly as possible is known to lead to good generalization in many situations. We will clarify the usage of the word "memorization" in the camera-ready version. For concrete examples from the cited papers,  let's consider the following two:
>
> [A] Behnam Neyshabur, Zhiyuan Li, Srinadh Bhojanapalli, Yann LeCun,Nathan Srebro. Towards understanding the role of over-parametrization in generalization of neural networks, 2018. URL: https://arxiv.org/pdf/1805.12076.pdf
>
> [B] Zitong Yang, Yu Bai, Song Mei. Exact gap between generalization error and uniform convergence in random feature models, 2020. URL: https://arxiv.org/pdf/2103.04554.pdf
>
> In the left of Figure 1 in A, we see that the test error is smallest when the training error is near zero (meaning that the entire training data has been memorized.) In the left of Figure 1 in B, the red curve is the maximum test error among models that interpolates the training data and the yellow curve is the test error found by the minimum norm interpolator. We see that the red curve (max test err. among memorizing models) captures the general trend of the yellow curve (test err. learned by ERM). These two figures provide complementary views supporting the claim that "memorization is sufficient for generalization [for neural networks, on real-world data]".
>
> > Intermediate layer representation
>
> For the ResNet-CIFAR, we simply use the pre-logit layer as the embedding (line 250). For language modeling, we use the last layers pre-MLP, post LayerNorm embedding (line 277).
>
> > Can you present some comparisons about space usage? For instance, in in line 257, you compare the computation of ResNet-14 to that of ResMem (on ResNet-8). What is the footprint of these two models?
>
> For a batch size of 1 and images of size 32 x 32, a ResNet\-8 (\~68K params) requires 2.5MB, while a ResNet-14 (\~128K params) requires 4MB. Embeddings from a ResNet\-8 and ResNet\-14 are both 64 dimensional. To embed the entire CIFAR100 training set (50K examples) requires ~15MB of disk space.
>
>
> > Practicality of ResMem hindered by additional memory cost
>
> We would like to propose ResMem both as
>    - (a) a conceptual idea that fits into the memorization/generalization puzzle that's been discussed in the literature, and at the same time
>    - (b) as a potentially practical algorithm; this can be engineered with approximate [1] or distributed [2, 3] nearest neighbor search algorithms as an efficient add-on to the original prediction model.
>
> We acknowledge that the representations of the training data could consume a large amount of disk space. However, we would like to point out that accelerators are in general several orders of magnitude more expensive than storage. We believe it'll be worthwhile to trade more space for less time spent on inferencing the accelerators.
>
> [1] Ruiqi Guo, Philip Sun, Erik Lindgren, Quan Geng, David Simcha, Felix Chern, and Sanjiv Kumar. Accelerating large-scale inference with anisotropic vector quantization. In International Conference on Machine Learning, 2020. URL https://arxiv.org/abs/1908.10396.
>
> [2] Jeff Johnson, Matthijs Douze, Herve Jegou. Billion-scale similarity search with GPUs, 2017. URL https://arxiv.org/pdf/2305.18466.pdf
>
> [3] Moritz Hardt, Yu Sun. Test-Time Training on nearest neighbors for large language models, 2023. URL https://arxiv.org/pdf/2305.18466.pdf

---

> > ### Comment · Reviewer_M8UD · 2023-08-12
> >
> > Thank you for your reply and additional details.
> >
> > On "memorization is sufficient for generalization," I would say that the phrase is precise but wrong. I might suggest something like "neural networks memorize and generalize." Even when restricted to standard neural networks on standard data sets, it's far from clear that memorization suffices for generalization: see [a], which finds "bad" global minima with zero train error but test error over 40%.
> >
> > [a] Liu, Shengchao, Dimitris Papailiopoulos, and Dimitris Achlioptas. "Bad global minima exist and sgd can reach them." Advances in Neural Information Processing Systems 33 (2020): 8543-8552.

---

> > > ### Author Response · Authors · 2023-08-14
> > > **Response to title "memorization is sufficient for generalization"**
> > >
> > > Thanks for sharing the reference. We agree that just having SGD-trained neural networks with 100% training accuracy doesn't guarantee generalization. We are more saying SGD trained memorization can *generally* leads to generalization. On consideration, we understand how the wording "sufficient" may be confusing. We are happy to change the wording to "Memorization is compatible with generalization". This is consistent with the discussion in our text.

---

### Official Review · Reviewer_1sN3 · 2023-07-07

**Soundness:** 3 good
**Presentation:** 3 good
**Contribution:** 3 good
**Rating:** 6
**Confidence:** 4

**Summary:**

The paper proposes the residual-memorization (ResMem) algorithm.
This algorithm aims to improve the generalization performance by explicitly memorizing the residual between the neural network's outputs and the training labels with a kNN regressor.
The paper demonstrates the validity of the algorithm from both the theoretical and the empirical perspectives.
The main theoretical result shows the convergence rate of ResMem.
The empirical results show the performance increase of ResMem compared with standard neural networks on image and language datasets.

**Strengths:**

The proposed method is quite novel.
The writing is clear.
The paper consists of both theoretical and empirical justification.
This paper provides an additional perspective on improving generalization in neural networks.

**Weaknesses:**

While the method proposed in this paper is novel and the empirical results look promising, I am concerned about whether the theoretical result justifies the message that the proposed method improves model generalization via memorization.

On the theoretical result, the paper does show the asymptotic behavior of the prediction error of ResMem.
However, it is not clear if the asymptotic behavior shows that it improves model generalization, since the paper does not include the comparison with the case without ResMem.
In particular, the paper claims in Section 3.3 that $T_2$ captures an irreducible error of the risk that is in general not asymptotically zero;
however, it is known that for the linear model with ERM investigated in this paper in the overparameterized regime, the generalization error is asymptotically zero under a range of certain circumstances.

**Questions:**

I am happy to raise the score if the authors address my concern.

Q1: In Section 3.2, why is the equality in the second line of equation under Line 163 correct and consistent with the first line?

Q2: In Section 3.3, what does the $T_2$ part of error represent?
The author claims that $T_2$ arises due to the limited capacity of $F$, and captures an irreducible error of the risk.
Could the authors please justify the argument?
Can this $T_2$ term represent the generalization error?

Q3: In Section 3.4, could the authors compare the bound with the case without ResMem?
In particular, what is the error difference with or without ResMem in the $T_2$ term?

Q4: On the empirical result, what is the additional computation and memory cost of the ResMem algorithm?
In particular, how do the additional memory cost, training time and estimation time of the kNN component compare with those of the DNN component?

Minor issues & suggestions:
- Typo: In Line 67, "mall model"
- The training error comparison in Figure 2 does not seem to be a fair comparison when the proposed ResMem guarantees a near 100% training accuracy. The test error comparison is the more informative one.

---
Thank you for the clarification, especially on clarifying Assumption 3.2.
I updated the score accordingly.

**Limitations:**

Does not apply

---

> ### Author Rebuttal · Authors · 2023-08-03
>
> We thank the reviewer for seeing the novelty of ResMem algorithm. We will fix the minor issues and adopt the suggestions.
> We address Q1-Q4 below.
>
> > ReQ1: equation under line 163.
>
> We use the linearity of expectation and use the fact that
> $\mathbb{E}\_{\mathbf{x}\sim\mathbb{P}\_{\mathbf{x}}} [\mathbf{x}\mathbf{x}^\mathsf{T}]=\mathbf{I}$
> from Assumption 3.1.
>
> $$
> \arg\min\_{\Vert\mathbf{\theta}\Vert\leq L} \mathbb{E}\_{\mathbf{x}\sim\mathbb{P}\_\mathbf{x}} [\langle\mathbf{\theta}-\mathbf{\theta}\_\star, \mathbf{x}\rangle]^2,
> $$
>
> $$
> =\arg\min\_{\Vert\mathbf{\theta}\Vert\leq L} \mathbb{E}\_{\mathbf{x}\sim\mathbb{P}\_\mathbf{x}} [(\mathbf{\theta}-\mathbf{\theta}\_\star)^\mathsf{T} \mathbf{x}\mathbf{x}^\mathsf{T}(\mathbf{\theta}-\mathbf{\theta}\_\star)^\mathsf{T}\rangle],
> $$
>
> $$
> =\arg\min\_{\Vert\mathbf{\theta}\Vert\leq L} \mathbb{E}\_{\mathbf{x}\sim\mathbb{P}\_\mathbf{x}} [(\mathbf{\theta}-\mathbf{\theta}\_\star)^\mathsf{T} \mathbf{x}\mathbf{x}^\mathsf{T}(\mathbf{\theta}-\mathbf{\theta}\_\star)^\mathsf{T}],
> $$
>
> $$
> =\arg\min\_{\Vert\mathbf{\theta}\Vert\leq L} (\mathbf{\theta}-\mathbf{\theta}\_\star)^\mathsf{T} \mathbb{E}\_{\mathbf{x}\sim\mathbb{P}\_\mathbf{x}} [\mathbf{x}\mathbf{x}^\mathsf{T}] (\mathbf{\theta}-\mathbf{\theta}\_\star)^\mathsf{T},
> $$
>
> $$
> =\arg\min_{\Vert\mathbf{\theta}\Vert\leq L} \Vert\mathbf{\theta}-\mathbf{\theta}\_\star\Vert^2
> $$
>
> $$
> =\arg\min_{\Vert\mathbf{\theta}\Vert\leq L} \Vert\mathbf{\theta}-\mathbf{\theta}\_\star\Vert = L\mathbb{\theta}\_\star
> $$
>
> The last equality comes from the fact that the projection of the vector $\mathbf{\theta}\_\star$ with norm $\Vert\mathbf{\theta}\_\star\Vert=1$  onto a norm ball of radius $L<1$ (Assumption 3.2) is simply the rescaled $L\cdot\mathbf{\theta}\_\star$.
>
> > ReQ2: $T_2$ at line 175.
>
> $T\_2$ alone doesn't represent generalization error. The sum of $T\_1$ and $T\_2$ (c.f. line 175) constitutes an upper bound for the generalization error.  Looking at the the expression for $T\_2$ under line 175, the term stems from the difference between $f\_\infty$ and $f\_\star$. As discussed in line 162-168, $f\_\infty(\mathbf{x})=\langle\mathbf{\theta}\_\infty, \mathbf{x}\rangle$ minimizes the population error  (ERM with infinite samples. This is why we describe it as "irreducible error of risk"). In this case, we would have $f\_\infty=f\_\star$ if the function class is large enough that $f\_\star\in\mathcal{F}$. However, because of the Assumption 3.2 that we imposed, we don't have $f\_\star\in\mathcal{F}$, so $T_2$ is not asymptotically zero. This justifies the claim that $T_2$ appears due to limited capacity of $\mathcal{F}$. Assumption 3.2 is precisely what distinguishes our conclusion from most papers that analyzes linear regression risk.
>
>
>
> > ReQ3: bound without ResMem.
>
> We first acknowledge that this is a good comparison and we will add to the camera-ready version. Looking at the generalization error under line 190, without ResMem, $T_1\lesssim d^2 L^2 n^{-2/3}$ would stay the same modulo minor change in constants. In contrast, $T_2$ would be independent of $n$: $T_2\lesssim d^2(1-L)^2$. Therefore, when $n$ is large, the risk with ResMem would be smaller.
>
> > ReQ4: computation and memory cost in empirical experiments.
>
> We discuss the computation cost for image classification experiment between line 257 and 260.
>
> """
>
> Computationally, we estimate the CPU latency of a CIFAR-ResNet-8 to be 15.9 ms for a single
> test image. By contrast, the k-NN step takes 4.8 ms for the same test image. To contextualize the
> latency cost, the total cost of ResMem with ResNet-8 (15.9 ms + 4.8 ms) is lower than the cost of the
> next-sized model, i.e., ResNet-14 (26.2 ms).
>
> """
>
> Regarding the memory cost, for a batch size of 1 and images of size 32 x 32, a ResNet-8 (\~68K params) requires 2.5MB, while a ResNet-14 (\~128K params) requires 4MB. Embeddings from a ResNet-8 and ResNet-14 are both 64 dimensional. To embed the entire CIFAR100 training set (50K examples) requires \~15MB of disk space. We will add this discussion on top of the current line 257-260.
>
>
> For the language experiment, the index set is quite large (1.6 billion tokens) and exact k-nearest neighbor search is infeasible. So we use the approximate nearest neighbor search algorithm ScaNN [1] (discussed in line 29, 280) to cut the kNN compute time.
>
> [1] Ruiqi Guo, Philip Sun, Erik Lindgren, Quan Geng, David Simcha, Felix Chern, and Sanjiv Kumar. Accelerating large-scale inference with anisotropic vector quantization. In International Conference on Machine Learning, 2020. URL https://arxiv.org/abs/1908.10396.

---

> > ### Comment · Reviewer_1sN3 · 2023-08-20
> >
> > Thank you for the clarification, especially on clarifying Assumption 3.2.
> > I updated the score accordingly.

---

### Decision · Program_Chairs · 2023-09-21

**Decision:**

Accept (poster)

**Comment:**

Four knowledgeable reviewers recommended accepting the paper: Borderline Accept, Weak Accept, Weak Accept, and Weak Accept. Based on the clear positive consensus, it is our recommendation to accept the paper. Authors should attend to the main points in the reviews. when preparing a final version. No basis to overturn the reviews.

The method proposed in this paper is novel, the empirical results are indeed good and supported by theory. This paper is valuable and should be shared within the community to advance research on ANN.